# G-Merging: Graph Model Merging for Parameter-Efficient Multi-Task Knowledge Consolidation

**Jun Chen**[1,2]  **Ziyue Qiao**[2,†]  **Qin Zhang**[1,†]  **Kaize Ding**[3]  **Xiao Luo**[4]

[1]College of Computer Science and Software Engineering, Shenzhen University
[2]School of Computing and Information Technology, Great Bay University
[3]Department of Statistics and Data Science, Northwestern University
[4]Department of Statistics, University of Wisconsin-Madison
242433011@stu.gbu.edu.cn
zyqiao@gbu.edu.cn
qinzhang@szu.edu.cn
kaize.ding@northwestern.edu
xiao.luo@wisc.edu
[†] Corresponding authors

## Abstract

The pretrain-finetuning paradigm has achieved notable success in graph learning. Moreover, merging models fine-tuned on different tasks to enable a parameter-efficient model with multi-task capabilities is gaining increasing attention for its practicality. However, existing model merging methods, such as weight averaging and task arithmetic, struggle to generalize well to graph structures and Graph Neural Network (GNN) models due to the unique structural heterogeneity of graph data. In this paper, we propose an innovative graph model merging framework called **G-Merging** for merging multiple task-specific fine-tuned GNN models. G-Merging first employs task arithmetic to coarsely merge graph models, capturing shared cross-task knowledge. Second, it introduces a Topology-aware Wasserstein Distance (TWD) loss to train lightweight task adapters upon the merged model, preserving domain-specific graph patterns via aligning the embeddings of merged and fine-tuned models. Third, G-Merging integrates the adapters into a training-free, topology-aware router within a mixture-of-experts (MoE) architecture, dynamically routing input graphs to task-specific adapters based on structural similarity, thereby mitigating conflicts and enhancing knowledge sharing. Extensive experiments on 8 graph downstream datasets demonstrate the effectiveness of the merged model, showing impressive performance close to or exceeding individual finetuned models while improving parameters and training efficiency. Our code is available at https://github.com/cjcj46262/G-Merging.

## 1 Introduction

With the gradual development of graph learning, various model architectures have been proposed (Scarselli et al., 2008; Wu et al., 2020; Xu et al., 2018; Wu et al., 2022), especially Graph Neural Networks (GNNs). Pre-trained models and pre-training strategies (Hu et al., 2020; Qiu et al., 2020; Xu et al., 2018; Gu et al.; Ju et al., 2024) on graph data have gained attention due to their strong generalization ability. Meanwhile, fine-tuning these pre-trained models on downstream tasks has become a standard paradigm (Sun et al., 2024; Zhang et al., 2022; Zhili et al., 2024; Sun et al., 2022; Gu et al., 2025), particularly in scenarios where data labels are limited or out-of-distribution, such as in the domains of chemistry (Kim et al., 2023; Mayr et al., 2018; Wu et al., 2018; Sterling & Irwin, 2015) or biology (Veličković et al., 2018; Ingraham et al., 2019; Zitnik et al., 2019). Fine-tuned models usually achieve good performance on specific downstream tasks, as they are trained on task-specific datasets in a targeted manner. However, when we need to perform multiple tasks simultaneously, applying individual fine-tuned models to different tasks results in high storage and

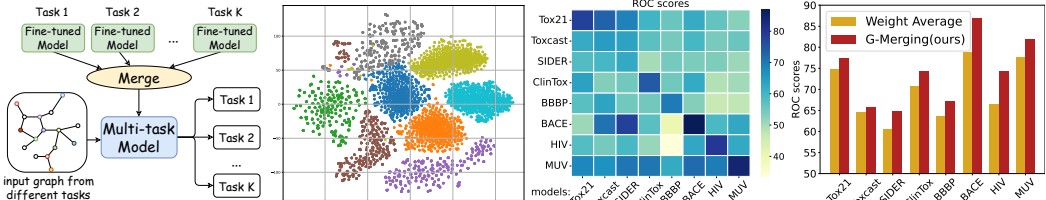

Figure 1: From left to right: (1) The illustration of the graph model merging task. (2) A 2D t-SNE visualization of the final-layer embeddings from eight molecular graph datasets, encoded using the graph pre-training model (Hu et al., 2020). (3) The cross-domain validation performance of task-specific graph models, each fine-tuned on one of the eight datasets from a shared pre-trained model. (4) The performance comparison between models merged via the Weight Average and our proposed methods.

deployment costs. Multi-Task Learning (MTL) offers a way to train a unified model on an aggregated multi-task dataset, but it usually learns from scratch and does not leverage the knowledge already encapsulated in fine-tuned models.

Thus, model merging seeks to construct a unified multi-task model by combining the parameters of fine-tuned models instead of joint training from scratch, as shown in Figure 1 (1). Ideally, it can retain the performance of each fine-tuned model while reducing parameter overhead and avoiding the need for extensive re-training on all task data. Until now, multiple model merging methods have been proposed, showing both practical significance and wide applicability. These range from the simple weight averaging strategy (Wortsman et al., 2022) to advanced ones, including Task Arithmetic (Ilharco et al., 2022), Ties-Merging (Yadav et al., 2023), AdaMerging (Yang et al., 2023), DARE (Yu et al., 2024), and Twin-Merging (Lu et al.). While model merging has been explored extensively in domains like vision and language models, its applicability in the context of graph models still remains a challenge. For example, as shown in Figure 1 (2), embeddings from eight similar graph datasets exhibit distinct clustering patterns, and in Figure 1 (3), the models from the same pre-training graph model, fine-tuned on one of these domains (row), usually fail to generalize to others (column) when replacing the backbones of the corresponding fine-tuning models. These observations indicate that graphs, inherently with heterogeneous structural patterns, lead to complex representations that are highly domain-specific. These can result in a significant performance gap between fine-tuned models and their merged model when applying existing merging strategies (Yang et al., 2024a; Huang et al., 2025), due to task conflict and knowledge sparsity (Yadav et al., 2023).

These observations motivate the need for novel model merging strategies tailored to graph data. In this paper, we propose an innovative framework called G-Merging for merging multi-task fine-tuned GNN models. Specifically, we first conduct a coarse merging on multiple fine-tuned GNN models via task arithmetic to obtain a unified GNN model. Then, to capture nuanced task-specific knowledge, we introduce the Topology-aware Wasserstein Distance (TWD) as a feature alignment loss. This loss trains lightweight, task-specific adapters upon the merged model, which supplement knowledge and alleviate representation bias. Furthermore, to mitigate task conflicts and encourage inter-task knowledge sharing, we further integrate these adapters into a Mixture of Experts (MoE) framework with a training-free, topology-aware router, which, together with the merged model, forms the final multi-task model to handle various downstream tasks. We demonstrate the effectiveness and generalization ability of G-Merging through extensive experiments on eight graph datasets, encompassing various GNN architectures and pre-training strategies. As an experimental example shown in Figure 1 (4), our method achieves a clear performance boost over naive weight averaging. Meanwhile, our approach offers efficient storage and fast inference, requiring storage space nearly equivalent to a single GNN model. The main contributions can be summarized as follows:

❶ This paper proposes G-Merging, a novel approach to merging fine-tuned graph models via *task arithmetic* and *TWD-based adapter routing*, resolving cross-domain structural heterogeneity and consolidating task-specific knowledge while enabling cross-task knowledge sharing.

❷ This paper proposes a *topology-aware* and *training-free* MoE that dynamically selects adapters at inference, enabling efficient cross-task knowledge transfer and multi-task generalization.

❸ Extensive experiments demonstrate that G-Merging not only *maintains or exceeds* the performance of individual fine-tuned models but also improves *storage and training efficiency*. The framework is also *model-agnostic*, supporting integration with various graph models.

## 2 RELATED WORK

**Model Merging.** In recent years, model merging has become a rapidly evolving technique applied in two dominant scenarios: (i) merging multiple models trained on the same dataset, aiming to improve model generalization (Wortsman et al., 2022; Wang et al., 2022; Cha et al., 2021; Gupta et al., 2020; Wang et al., 2025) or support federated learning (Liu et al., 2022a; Wang et al., 2020) (ii) merging multiple models trained on different datasets or for different tasks to perform MTL (Matena & Raffel; Jin et al., 2022; Yang et al., 2023; Huang et al., 2025). This paper primarily focuses on the latter scenarios. Consequently, numerous works propose advanced merging techniques to improve performance or efficiency (Huang et al., 2025; Yadav et al., 2023; Lu et al.; Ilharco et al., 2022). The baselines in our experiments provide specific examples.

**Fine-tuning in Graph Transfer learning.** Supervised Fine-tuning (SFT) from pre-trained models on downstream tasks is becoming a standard paradigm in both NLP and vision fields (Paul & Chen, 2022; Dodge et al., 2020; Devlin et al., 2019; Bommasani et al., 2021; Dosovitskiy et al., 2020; Ma et al.; Qiao et al., 2023) and is gaining increasing popularity in graph learning. SFT in graph transfer learning demonstrates effective performance while alleviating the burden of collecting labels regarding new tasks. Depending on whether all the parameters of models are adjusted, SFT can be divided into conventional full fine-tuning (FFT) and parameter-efficient fine-tuning (PEFT). PEFT aims to reduce the number of trainable parameters for downstream tasks by either inserting additional modules (e.g. adapters, learnable prompts) or training only a subset of parameters while freezing the rest (Hu et al.; Liu et al., 2022b; Houlsby et al., 2019a).

**Mixture of Experts.** The Mixture of Experts (MoE) paradigm introduces an adaptive routing method that allocates experts dynamically to handle different inputs. Sparse MoE models activate only a subset of experts for each input to improve computational efficiency, while dense MoE models combine the outputs of all experts to achieve superior performance. The concept was first introduced by (Jacobs et al., 1991) with gating mechanisms to select the experts. Recent studies have focused on challenges such as load balancing of experts (Clark et al., 2022; Zhou et al., 2022), training instability (Zoph et al., 2022), expert specialization (Dai et al., 2024; Tang et al., 2024), and synchronization reduction (Sukhbaatar et al., 2024) for tasks in CV and NLP fields. Despite their success, the requirement for substantial multi-task data and the high cost of joint training remain significant challenges for these methods. In contrast, our MoE module is completely train-free and orchestrates task-specific experts based on TWD.

## 3 PRELIMINARIES

**Notation.** Let $G(\mathcal{V}, \mathcal{E})$ be a graph with vertices $\mathcal{V}$ and edges $\mathcal{E}$. An input graph can be expressed as $G = \{\mathbf{A}, \mathbf{X}\}$, where $\mathbf{X} \in \mathbb{R}^{|\mathcal{V}| \times d_{node}}$ is the node feature matrix and $\mathbf{A} \in \mathbb{R}^{|\mathcal{V}| \times |\mathcal{V}|}$ is the adjacency matrix. $\mathbf{A}_{ij} = 1$ otherwise 0 if there is an edge between nodes $n_i$ and $n_j$. Under the pretrain-finetune paradigm, let $f_\theta(G_i)$ be a pre-training GNN model with parameter $\theta_{pre}$, which is pre-trained on a large, general-purpose graph dataset. Then, we consider a collection of $K$ downstream graph tasks, indexed by $k = \{1, 2, ..., K\}$. Each task $k$ is associated with a private dataset $\mathcal{D}_k = \{(G_i, y_i)\}_{i=1}^{N_k}$, where $y_i$ is the label of the graph $G_i$ and $N_k$ is the number of samples in task $k$. For each task, the model is initialized with $\theta_{pre}$ and then fine-tuned on $\mathcal{D}_k$ to obtain task-specific parameters $\theta_k$ with an additional task-specific prediction head.

**Problem Definition.** The graph model merging problem aims to obtain a unified graph model $f_{\theta_m}$ with reduced parameter overhead by consolidating knowledge from a collection of fine-tuned models $\{f_{\theta_1}, f_{\theta_2}, ..., f_{\theta_K}\}$ and pre-trained model $f_{\theta_{pre}}$, eliminating the need to maintain a full set of parameters for each task-specific model. Then, $f_{\theta_m}$ as a shared graph encoder, combined with the task-specific classification heads from the fine-tuned models, is used to perform multi-task inference. The objective is to achieve this without retraining a new multi-task model from scratch, which can be computationally expensive and require access to all labeled data, or relying on naive parameter averaging strategies, which often result in degraded performance due to parameter misalignment and lack of task-specific nuance. The goal is to strike a balance by efficiently merging multiple task-specific fine-tuned models into a unified model that *reduces storage and computational cost* by

eliminating the overall number of parameters, *maintains or even improves individual performance* on all downstream tasks, and *possibly leverages shared knowledge* to enable cross-task generalization.

**Wasserstein Distance.** Wasserstein Distance (WD) (Peyré et al., 2019)(a.k.a. Earth Mover's Distance, or Optimal Transport Distance) quantifies the similarities between objects such as probability distributions, either discrete or continuous, by computing the minimum cost of an optimal transport plan from one to the other (Bécigneul et al., 2020). We now describe the definition of WD between two discrete distributions as follows.

**Definition 1.** *Consider two discrete probability measures $\mu \in P(\mathbb{X})$ and $\nu \in P(\mathbb{Y})$, represented respectively as $\mu = \sum_{i=1}^{n} \mathbf{u}_i \delta_{x_i}$ and $\nu = \sum_{j=1}^{m} \mathbf{v}_j \delta_{y_j}$, where $\delta_x$ denotes the Dirac measure centered at $x$. A coupling of $\mu(x)$ and $\nu(y)$ can be expressed as $\tau(x, y) = \sum_{i=1}^{n} \sum_{j=1}^{m} \mathbf{T}_{ij} \delta_{(x_i, y_j)}$, in which $\mathbf{T} \in \mathbb{R}_+^{n \times m}$ fulfills the marginal conditions $\mathbf{T}\mathbf{1}_n = \mathbf{u}$ and $\mathbf{T}^\top \mathbf{1}_m = \mathbf{v}$. Here, $\mathbf{u}$ and $\mathbf{v}$ are the respective weight vectors of $\mu$ and $\nu$, and $\mathbf{1}_n \in \mathbb{R}^n$ is the all-ones vector of length $n$. The Wasserstein distance between $\mu$ and $\nu$ is then given by:*

$$\mathcal{D}_{wd}(\mu, \nu) = \min_{\mathbf{T} \in \Pi(\mathbf{u}, \mathbf{v})} \sum_{i=1}^{n} \sum_{j=1}^{m} \mathbf{T}_{ij} \cdot c(x_i, y_j). \tag{1}$$

*where the feasible set $\Pi(\mathbf{u}, \mathbf{v}) = \{\mathbf{T} \in \mathbb{R}_+^{n \times m} \mid \mathbf{T}\mathbf{1}_n = \mathbf{u} \wedge \mathbf{T}^\top \mathbf{1}_m = \mathbf{v}\}$ includes all joint distributions with marginals $\mu$ and $\nu$. The function $c(x_i, y_j)$ denotes the ground cost associated with transporting mass from $x_i$ to $y_j$. The matrix $\mathbf{T}$, referred to as the **transport plan** or **transport map**, defines the quantity $\mathbf{T}_{ij}$ of mass relocated from $x_i$ to $y_j$.*

## 4 METHODOLOGY

### 4.1 COARSE PARAMETERS MERGING VIA TASK ARITHMETIC

We begin with a coarse merging process to reduce parameter redundancy. Specifically, from previous investigations and discussions on model merging research, *knowledge modularization is an effective and reasonable technique that decomposes the knowledge possessed by experts into* ❶ *Shared knowledge and* ❷ *Task-specific exclusive knowledge (Lu et al.; Huang et al., 2025).* As the name suggests, shared knowledge represents common and generalized knowledge across different tasks, *e.g.*, in the pretrain-finetune paradigm, the base model or pre-trained model possesses shared knowledge across downstream tasks. Furthermore, task-specific knowledge can be compressed into shared knowledge by merging the parameters of fine-tuned models (*e.g.*, direct weight averaging is a simple approach). Based on the above observation, we merge fine-tuned GNN models into a unified model with shared knowledge, leveraging an established merging technique called Task Arithmetic (Ilharco et al., 2022).

Specifically, for $K$ tasks, the corresponding task vectors of parameters are defined as $\tau_k = \theta_k - \theta_{pre}$, where $k \in \{1, 2, ..., K\}$. Furthermore, multiple task vectors $\{\tau_k\}_{k=1}^{K}$ are added and merged into the pre-trained parameters $\theta_{pre}$, formulated as $\theta_{uni} = \theta_{pre} + \lambda \sum_{k=1}^{K} \tau_k$, where the $\theta_{uni}$ is the parameters of unified model with

---

**Algorithm 1** G-Merging

**Require:** fine-tuned model $\{f_{\theta_1}, f_{\theta_2}, ..., f_{\theta_K}\}$, pre-trained model $f_{\theta_{pre}}$, loss function $\mathcal{L}_{TWD}$ $\mathcal{L}_{MD}$, non-parametric router $\mathcal{R}$, and pre-specified weight $\lambda$

1: **Parameters merging:** ▷ Shared knowledge
2: Compute the unified model $f_{\theta_{uni}}$:
3:     $\theta_{uni} \leftarrow \theta_{pre} + \lambda \sum_{k=1}^{K} (\theta_k - \theta_{pre})$

4: **Training:** ▷ Exclusive knowledge
5: **for** each task $k$ **do**
6:     Transfer knowledge of $f_{\theta_k}$ into task-specific adapters $f_{adap, \theta_k^*}$:
7:       $\theta_k^* \leftarrow \text{train}\{f_{\theta_k}, f_{\theta_{uni}}, \mathcal{L}_{TWD}, \mathcal{L}_{MD}\}$
8: **end for**

9: **Inference:** ▷ Main inference loop
10: perform task $k$
11: input a graph $G$ with embedding matrix $\mathbf{H}$
12: **for** each layer in model **do**
13:     Update embeddings by unified model:
14:       $\mathbf{H} \leftarrow f_{\theta_{uni}}(\mathbf{H})$
15:     Calculate router weights:
16:       $\{w_k\}_{k=1}^{K} \leftarrow \mathcal{R}(\{f_{adap, \theta_k^*}(\mathbf{H})\}_{k=1}^{K}, k)$
17:     Merge for final output $\mathbf{H}^{(l)}$:
18:       $\mathbf{H} \leftarrow \mathbf{H} - \sum_{k=1}^{K} w_k \cdot f_{adap, \theta_k^*}(\mathbf{H})$
19: **end for**

**Ensure:** Output $\mathbf{H}$ for input graph $G$.

---

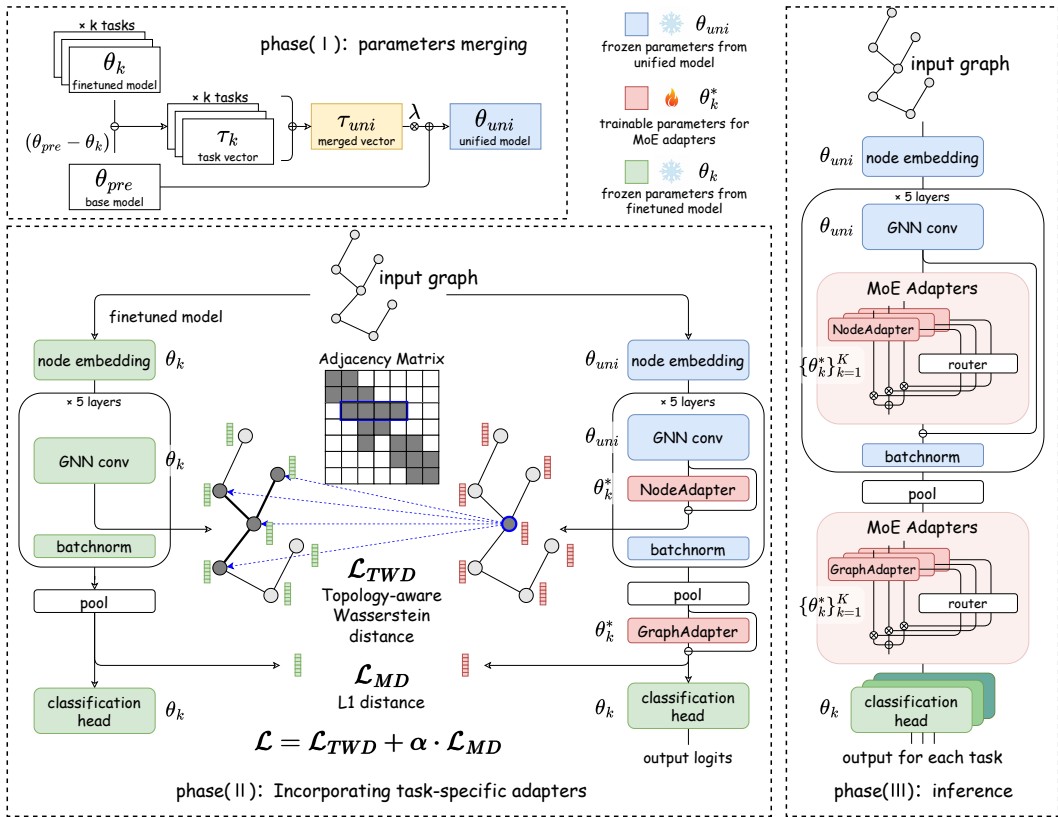

Figure 2: Overall framework of G-Merging, containing three main phases. In phase(I), we coarsely merge GNN models fine-tuned on different tasks into a unified model using task arithmetic. In phase(II), we train multiple task-specific adapters, lightweight modules used to solve the representation bias between the unified model and the fine-tuned model. Moreover, we apply the graph Topology-Aware Wasserstein Distance (denoted as $\mathcal{L}_{TWD}$ in the figure) and the L1 distance (denoted as $\mathcal{L}_{MD}$ in the figure) to promote representation alignment at the node and graph levels, respectively. In Phase (III), we add MoE adapters, composed of task-specific adapters and a router method, at each layer of the unified GNN model and before the prediction head for test-time inference.

shared knowledge and $\lambda$ is a scaling hyperpa-
rameter to control the balance between fundamental knowledge in pre-trained model and task-related knowledge in fine-tuned models. Setting $\lambda$ as $\frac{1}{K}$ recovers naive weight averaging, while larger/smaller values amplify or suppress task-related adjustments. From the empirical results in Appendix F, a value bigger than $\frac{1}{K}$ may be more effective for downstream tasks. As shown in Phase(I) of Figure 2, this coarse merging process offers a foundational backbone that captures commonalities across tasks, serving as an initialization for the subsequent process.

## 4.2 INCORPORATING TASK-SPECIFIC ADAPTERS

While coarse merging provides a foundation for capturing shared knowledge, it may be insufficient for capturing nuanced, task-specific knowledge. Empirically, we notice that the single unified GNN model we obtained exhibits a performance gap compared to the fine-tuned GNN models on downstream graph tasks. Previous research has demonstrated that this discrepancy is attributed to representation bias, which refers to a substantial difference in the representation distribution between the unified and fine-tuned models (Yang et al., 2024a;b). To address this problem, we further introduce additional adapter-based modules, which are inserted after the graph convolution layers and before the classification head. We trained task-specific adapters to minimize representation bias at both the node level (i.e., NodeAdapters) and the graph level (i.e., GraphAdapters).

Specifically, given an input graph data $G = \{\mathbf{A}, \mathbf{X}\}$, the node embedding matrices $\mathbf{H}^{(l)} \in \mathbb{R}^{|\mathcal{V}| \times d}$ ($\mathbf{H}^{(0)} = \mathbf{X}$) updated in $l$-th GNN layers with or without NodeAdapters are respectively formulated as

$$\mathbf{H}_{\theta,\theta^*}^{(l)} = f_{conv,\theta}^{(l)}(\mathbf{A}, \mathbf{H}_{\theta,\theta^*}^{(l-1)}) - f_{adap,\theta^*}(f_{conv,\theta}^{(l)}(\mathbf{A}, \mathbf{H}_{\theta,\theta^*}^{(l-1)})),$$
$$\mathbf{H}_\theta^{(l)} = f_{conv,\theta}(\mathbf{A}, \mathbf{H}_\theta^{(l-1)}),$$
(2)

where $\theta$ and $\theta^*$ denote the parameters of GNN model and adapters, respectively, $f_{conv}$ is the graph convolution function for aggregating messages and updating embeddings, which varies with different GNN backbones like GIN or GCN, and $f_{adap}$ is the lightweight adapter module, which can be an arbitrary implementation (such as multiple fully connected layers). Without loss of generality, in our method, we follow the original work on Adapters (Houlsby et al., 2019b) and set $f_{adap}$ as follows:

$$f_{adap,\theta^*}(\mathbf{H}) = \text{ReLU}(\mathbf{H} \cdot \mathbf{W}_{down}) \cdot \mathbf{W}_{up},$$
(3)

where $\mathbf{W}_{down} \in \mathbb{R}^{d \times r}$ and $\mathbf{W}_{up} \in \mathbb{R}^{r \times d}$ are two trainable matrices, $i.e.$, $\theta^* = \{\mathbf{W}_{down}, \mathbf{W}_{up}\}$, ReLU$(\cdot)$ is a nonlinear activation function. The parameter $r$ represents the internal rank of the adapters, controlling the scale. Similarly, the final graph embedding vectors $\mathbf{h} \in \mathbb{R}^d$ extracted by the Pooling layer with or without GraphAdapter can be formulated as

$$\mathbf{h}_{\theta,\theta^*} = f_r(\{\mathbf{H}_\theta^{(l)}\}) - f_{adap,\theta^*}(f_r(\{\mathbf{H}_\theta^{(l)}\})),$$
$$\mathbf{h}_\theta = f_r(\{\mathbf{H}_\theta^{(l)}\}),$$
(4)

where $f_r$ is a readout function such as averaging or summarizing the node embeddings.

For each task $k$, we denote the task-specific adapter parameters as $\theta_k^*$. We train these adapters on the corresponding task graph data to align the distribution of embeddings extracted by the merged and the previous fine-tuned GNN models. To achieve this, we employ Wasserstein Distance in Section 3 to quantify the similarity between two sets of node embeddings in a graph, which can be regarded as discrete distributions (Zhang et al., 2022; Chen et al., 2020). Motivated by the earlier observation of domain-specific structural patterns inherent in graph data, we further incorporate graph topology to enhance structural awareness. Specifically, given the node embeddings $\mathbf{H}_{\theta_{uni},\theta_k^*}^{(l)}$ encoded from the $l$-th layer of the previously derived unified model with NodeAdapters and $\mathbf{H}_{\theta_k}^{(l)}$ from the $k$-th fine-tuned model, we introduce an alignment loss based on Topology-aware Wasserstein Distance:

$$\mathcal{L}_{TWD} = \text{TWD}(\mathbf{H}_{\theta_{uni},\theta_k^*}^{(l)}, \mathbf{H}_{\theta_k}^{(l)}, \mathbf{A}) = \min_{\mathbf{T} \in \Pi(\mathbf{A})} \sum_{i=1}^{|\mathcal{V}|} \sum_{j=1}^{|\mathcal{V}|} \mathbf{T}_{ij} \cdot c(\mathbf{h}_i^{(l)}, \mathbf{h}_j'^{(l)}),$$
(5)

where $\Pi(\mathbf{A}) = \{\mathbf{T} \in \mathbb{R}_+^{|\mathcal{V}| \times |\mathcal{V}|} \mid \mathbf{T}\mathbf{1}_{|\mathcal{V}|} = \frac{1}{|\mathcal{V}|} \cdot \mathbf{1}_{|\mathcal{V}|} \wedge \mathbf{T}^\top \mathbf{1}_{|\mathcal{V}|} = \frac{1}{|\mathcal{V}|} \cdot \mathbf{1}_{|\mathcal{V}|} \wedge \mathbf{T} \odot (\mathbf{1}_{|\mathcal{V}| \times |\mathcal{V}|} - \mathbf{A}) = \mathbf{0}_{|\mathcal{V}| \times |\mathcal{V}|}\}$ is the set of transport plans, $\mathbf{h}_i^{(l)} = \mathbf{H}_{\theta_{uni},\theta_k^*}^{(l)}[i,:]$ and $\mathbf{h}_j'^{(l)} = \mathbf{H}_{\theta_k}^{(l)}[j,:]$ denote two embedding vectors, and $c(\cdot, \cdot)$ is the cost function. Compared to the original WD, the transport plan $\mathbf{T}$ is now constrained by the graph adjacency matrix $\mathbf{A}$. Therefore, this loss function is computationally aware of graph topology.

Notably, in our task, we calculate the cosine distance $c(\mathbf{a}, \mathbf{b}) = \frac{1}{2}(1 - \cos(\mathbf{a}, \mathbf{b}))$ as the cost function, following many prior works based on the application of Optimal Transport problems (Zhang et al., 2022; Xu et al., 2020; Chen et al., 2020). Additionally, we set $\mathbf{A}$ as the 1-hop adjacency matrix with self-loops, $i.e.$, $\mathbf{A}_{ij} = 1$ if and only if $i = j$ or there exists an edge between $i$ and $j$. Conceptually, reducing the cosine distance between $\mathbf{h}_i$ and any $\mathbf{h}_j'$, where $j \in \mathcal{N}(i) \cup \{i\}$ (with $\mathcal{N}(i)$ denotes the set of neighbors of node $i$), contributes to the reduction of TWD. This is intuitively reasonable because the embeddings of neighboring nodes extracted by GNNs are expected to be similar—a property often referred to as the smoothness of GNNs (Li et al., 2018). Similar to the standard WD, calculating TWD requires finding out the optimal transport plan $\mathbf{T}$. This problem has been extensively studied in prior work, with efficient solutions such as Sinkhorn algorithm variants (Cuturi, 2013; Peyré et al., 2019; Dvurechensky et al., 2018; Zhang et al., 2022). We provide a detailed introduction including theoretical background and computational complexity of TWD in Appendix A.

For the final graph embeddings $\mathbf{h}_{\theta_{uni},\theta_k^*}$ and $\mathbf{h}_{\theta_k}$, we define the alignment loss as Manhattan Distance (also called L1 distance) between them, $i.e.$, $\mathcal{L}_{MD} = \|\mathbf{h}_{\theta_{uni},\theta_k^*} - \mathbf{h}_{\theta_k}\|_1$. Combining the losses above, the final optimization problem is

$$\mathcal{L} = \min_{\theta_k^*} \frac{1}{|\mathcal{D}_k|} \sum_{G \in \mathcal{D}_k} \left( \alpha \cdot \mathcal{L}_{MD} + \sum_{l=1}^{L} \mathcal{L}_{TWD} \right)$$
(6)

where $\alpha$ is the hyperparameter to balance two losses.

As shown in Figure 2, in Phase(II), we train without task labels and obtain $K$ sets of adapters corresponding to $K$ tasks. These adapters serve as supplemental task-specific knowledge to the shared knowledge in the unified model and improve the performance on downstream tasks. Notably, our training process is efficient because the majority of parameters $\theta_{uni}$ are frozen while only parameters in adapters $\theta_k^*$ remain for training.

### 4.3 INFERENCE PROCEDURE

In the inference phase, we aim to obtain an enhanced model by integrating the knowledge in adapters from different tasks. Inspired by the perspective that knowledge from similar tasks can be mutually beneficial (Ilharco et al., 2022), we compose the task-specific adapters into MoEAdapters based on a parameter-free Mixture-of-Experts architecture (Cai et al., 2024) during inference. Specifically, we compose $f_{adap,\theta^*}$ to $f_{moe,\{\theta_1^*,\theta_2^*,...,\theta_K^*\}}$ in Equations 2 and 4. Inside the MoEAdapter, each expert is set as the task-specific adapter, well-trained for a particular task, and the output is a weighted sum of the outputs of the experts. The process can be formulated as

$$f_{moe,\{\theta_1^*,\theta_2^*,...,\theta_K^*\}}(\mathbf{H}) = \sum_{k=1}^{K} w_k \cdot f_{adap,\theta_k^*}(\mathbf{H}) \tag{7}$$

where $\{\theta_1^*,\theta_2^*,...,\theta_K^*\}$ are parameters trained in phase(II), $\{w_1,w_2,...,w_K\}$ are MoE weights calculated by a train-free router module based on the similarity between different tasks. Precisely, assume that we have node embeddings $\mathbf{H} \in \mathbb{R}^{|\mathcal{V}| \times d}$ from a given graph $G$ and obtain two outputs, $f_{adap,\theta_A^*}(\mathbf{H})$ and $f_{adap,\theta_B^*}(\mathbf{H})$, from two task-specific adapters. We can consider task A and task B to be similar if $f_{adap,\theta_A^*}(\mathbf{H})$ and $f_{adap,\theta_B^*}(\mathbf{H})$ are similar, and the similarity can be measured by TWD. On this basis, during inference on task k, $\{w_1,w_2,...,w_K\}$ are calculated as follows:

$$\{w_1,w_2,...,w_K\} = \text{softmax}(\{-\text{TWD}(f_{adap,\theta_i^*}(\mathbf{H}), f_{adap,\theta_k^*}(\mathbf{H}))\}_{i=1}^{K}) \tag{8}$$

This mechanism amplifies the contribution of experts trained on similar tasks when addressing the target task. Additionally, the similarity between two graph embeddings $\mathbf{h} \in \mathbb{R}^d$ can be measured by Manhattan Distance, and the $\{w_1,w_2,...,w_K\}$ on graph level are calculated as:

$$\{w_1,w_2,...,w_K\} = \text{softmax}(\{-\|f_{adap,\theta_i^*}(\mathbf{h}) - f_{adap,\theta_k^*}(\mathbf{h})\|_1\}_{i=1}^{K}) \tag{9}$$

In this way, the MoEAdapters are tailored for each graph instance and effectively utilize knowledge while mitigating knowledge conflict across different tasks. As a result, we only deploy $\theta_{uni}$ and $\{\theta_1^*,\theta_2^*,...,\theta_K^*\}$ during inference for all tasks, which is significantly smaller than the total size of $\theta_1,\theta_2,...,\theta_K$ from all finetuned models, according to the experimental data in Table 4.

## 5 EXPERIMENTS

### 5.1 EXPERIMENTAL SETUP

**Datasets.** We use 8 binary graph classification datasets for molecule property prediction as downstream tasks, which are widely used as benchmarks for evaluating pretrain-finetune strategy in previous work (Zhang et al., 2022; Kim et al., 2023; Zhili et al., 2024; Sun et al., 2024). Data statistics and preprocessing are detailed in Appendix B and C.

**Baselines.** Since we have not found related work in merging GNN models, we compare our G-Merging with several typical and advanced model merging methods from the CV or NLP field, including: Weight Averaging, Task Arithmetic (Ilharco et al., 2022), Ties-Merging (Yadav et al., 2023), and EMR-Merging (Huang et al., 2025), AdaMerging (Yang et al., 2023), Twin-merging (Lu et al.) (detailed in Appendix D). Furthermore, we include Multi-task Learning, individual fine-tuned models, and the pre-trained model as additional baselines beyond the merging strategy. The fine-tuned and pre-trained models serve as the upper and lower performance bounds, respectively.

Table 1: Test ROC-AUC score (%) of GIN models (contextpred) on downstream molecular property prediction tasks after merging fine-tuned models. (" $*$ " denotes performance surpassing that of the fine-tuned model.)

| Methods | Tox21 | Toxcast | SIDER | ClinTox | BBBP | BACE | HIV | MUV | Average |
|---|---|---|---|---|---|---|---|---|---|
| Full Fine-Tuned | 78.0 | 64.8 | 62.5 | 74.0 | 69.6 | 86.8 | 79.6 | 83.9 | 74.9 |
| Pretrained | 68.9 | 63.3 | 58.1 | 61.9 | 55.3 | 78.5 | 59.1 | 72.3 | 64.7 |
| Multi-Task Learning | 75.5 | 63.4 | 62.8 | 64.9 | 66.4 | 84.7 | 74.8 | 77.5 | 71.2 |
| Weight Average | 74.7 | 64.5 | 60.4 | 70.7 | 63.5 | 78.8 | 66.5 | 77.5 | 69.6 |
| Task Arithmetic | 74.2 | 64.6 | 60.4 | $77.6^*$ | 66.6 | 70.8 | 68.5 | 74.8 | 69.7 |
| Ties-Merging | 69.2 | 63.4 | 57.8 | 62.1 | 55.5 | 79.0 | 61.6 | 77.5 | 65.8 |
| EMR-Merging | 77.6 | 63.5 | 62.2 | 72.8 | **69.1** | 80.9 | 74.8 | 71.3 | 71.5 |
| AdaMerging | 69.2 | 62.0 | 57.6 | 64.6 | 60.2 | 70.5 | 64.2 | 66.7 | 64.4 |
| Twin-Merging | 69.9 | 63.4 | 59.0 | 63.0 | 59.0 | 57.5 | 59.8 | 75.7 | 63.4 |
| **G-Merging-s (Ours)** | $77.2_{\pm0.4}$ | $65.7^*_{\pm0.1}$ | $64.6^*_{\pm0.5}$ | $76.0^*_{\pm0.5}$ | $67.0_{\pm0.2}$ | $86.6_{\pm0.2}$ | $\mathbf{76.0}_{\pm0.3}$ | $80.8_{\pm0.5}$ | **74.2** |
| **G-Merging (Ours)** | $\mathbf{77.4}_{\pm0.5}$ | $\mathbf{65.8}^*_{\pm0.1}$ | $\mathbf{64.8}^*_{\pm0.6}$ | $74.2^*_{\pm0.6}$ | $67.1_{\pm0.2}$ | $\mathbf{86.8}_{\pm0.2}$ | $74.2_{\pm0.4}$ | $\mathbf{81.9}_{\pm0.5}$ | 74.0 |

Table 2: Test ROC-AUC score (%) of GIN models (edgepred) on downstream molecular property prediction tasks after merging fine-tuned models.

| Methods | Tox21 | Toxcast | SIDER | ClinTox | BBBP | BACE | HIV | MUV | Average |
|---|---|---|---|---|---|---|---|---|---|
| Full Fine-Tuned | 76.1 | 66.1 | 64.8 | 70.0 | 70.5 | 86.1 | 77.6 | 79.7 | 73.9 |
| Pretrained | 71.6 | 64.9 | 60.0 | 61.6 | 54.6 | 76.4 | 64.4 | 65.5 | 64.9 |
| Multi-Task Learning | 73.8 | 64.0 | 63.3 | 71.2 | 68.8 | 81.1 | 74.5 | 72.8 | 71.2 |
| Weight Average | 74.1 | **66.1** | 62.2 | 65.9 | 63.6 | 78.7 | 67.0 | 68.5 | 68.3 |
| Task Arithmetic | 74.1 | 65.8 | 63.1 | 71.6 | 66.1 | 75.7 | 68.9 | 66.8 | 69.0 |
| Ties-Merging | 71.7 | 65.0 | 59.7 | 61.5 | 54.8 | 78.3 | 64.4 | 66.6 | 65.2 |
| EMR-Merging | 76.6 | 65.3 | 64.1 | 67.8 | 70.4 | 82.1 | 70.2 | 67.1 | 70.4 |
| AdaMerging | 71.3 | 63.6 | 59.7 | 72.7 | 57.3 | 72.4 | 68.6 | 60.8 | 65.8 |
| Twin-Merging | 71.8 | 64.6 | 58.8 | 67.6 | 56.4 | 56.7 | 61.6 | 69.0 | 63.3 |
| **G-Merging-s (Ours)** | $77.3^*_{\pm0.4}$ | $66.0_{\pm0.1}$ | $64.6_{\pm0.2}$ | $\mathbf{74.1}^*_{\pm0.5}$ | $\mathbf{69.2}_{\pm0.3}$ | $83.1_{\pm0.6}$ | $\mathbf{74.9}_{\pm0.7}$ | $75.5_{\pm0.5}$ | **73.1** |
| **G-Merging (Ours)** | $76.9^*_{\pm0.4}$ | $66.0_{\pm0.1}$ | $\mathbf{64.8}_{\pm0.2}$ | $71.8^*_{\pm0.4}$ | $68.9_{\pm0.3}$ | $\mathbf{84.6}_{\pm0.5}$ | $74.2_{\pm0.7}$ | $\mathbf{77.4}_{\pm0.6}$ | **73.1** |

**Settings.** To evaluate the effectiveness of G-Merging in various scenarios, we use models with a range of GNN backbones. Specifically, we reuse the pretrained models provided by Weihua *et al.* (Hu et al., 2020) with two GNN architectures: GIN (Xu et al., 2018) and GCN (Kipf & Welling, 2016), and two pretrain strategies: *contextpred* and *edgepred*. All models are self-supervised and pretrained on the chemistry dataset ZINC15 (Sterling & Irwin, 2015) (containing over 2 million molecules). We obtain the fine-tuned models by fully fine-tuning pretrained models on 8 downstream tasks. We run our method with 5 different random seeds and report the mean and standard deviation of the performance. In addition, we also validate a simplified version of our method, G-Merging-s, which directly uses task-specific adapters but not MoEAdapters during inference. More details of pretrained models and hyperparameters are provided in Appendix B.

## 5.2 EXPERIMENTAL RESULTS AND ANALYSIS

**Main Results.** The ROC-AUC scores in Table 1, Table 2, and Table 6 (in Appendix E) show the results for all tasks with various pretrained GNN models. The total average scores across 8 datasets indicate the overall capability of merging methods. We observe that our method significantly outperforms the baseline method in the vast majority of tasks, regardless of the GNN architectures and pretraining strategies. The performance is competitive with finetuned models, and outperforms them in some tasks, indicating that our merging method *maintains or even improves individual performance*, which is perhaps attributed to knowledge transfer or knowledge complementarity across tasks. Comparing G-Merging with G-Merging-s, G-Merging can perform better on certain tasks, indicating that one task can benefit from others in the MoE structure. Additionally, for tasks with more pronounced topological heterogeneity, such as HIV and MUV (datasets with numerous and structurally complex molecules), our method achieves a more significant performance improvement over the baselines. This suggests that the TWD loss effectively extracts topological information from graph data, which is an essential element of the knowledge in the merged GNN model.

**Ablation Studies.** To examine the effect of each component on the final performance, we conduct ablation studies on eight downstream tasks using the GIN (contextpred) pretrained model. We design

five variants of G-Merging by removing certain components, and their overall performances are shown in Figure 3 (more results are shown in Appendix E Figure 6).

*Effect of parameter merging.* When we directly use the original pretrained model as the unified model ($\theta_{uni} = \theta_{pre}$) and skip the phase of parameter merging, the performance significantly drops. This suggests that shared knowledge is an indispensable fundamental part of a model before incorporating task-specific exclusive knowledge.

*Effect of MoE adapters.* The performance of the two variant methods without MoEAdapters at either the node or graph level is worse than that of G-Merging, and the model performs the worst when MoEAdapters are removed at the graph level. This demonstrates the importance of MoEAdapters as task-specific knowledge suppliers and their effectiveness in alleviating representation bias.

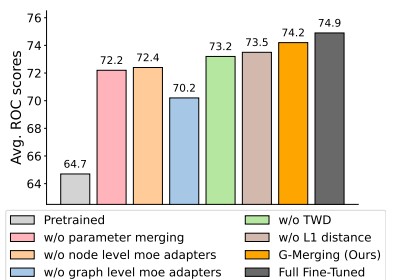

Table 3: Ablation Studies. Average ROC-AUC scores across 8 tasks for five ablated variants of G-Merging

*Effect of loss functions.* We further validate the effectiveness of $\mathcal{L}_{TWD}$ and $\mathcal{L}_{MD}$ by disabling one of them during training. From the results, it can be seen that $\mathcal{L}_{TWD}$ improves performance more significantly as it effectively leverages graph topological information. In addition, $\mathcal{L}_{MD}$ further boosts performance by enhancing graph embeddings alignment based on this.

**Routing Analysis.** Here we perform an analysis of the TWD-based router method in our MoE structure. We extract the average of expert weights during inference, shown in Figure 4 and Appendix E, Figure 5. It is evident that each sample receives maximum weights on its task expert adapters. Furthermore, the router tends to allocate a greater weight to expert adapters when there is higher similarity between the test task and the source task of the expert, and this relationship is bidirectional. For instance, considering the two tasks, ClinTox and SIDER, which focus on the toxicity of drug compounds and diverse adverse reactions, respectively, they intuitively share a high degree of similarity. Consequently, the router tends to allocate greater weights to expert_ClinTox when performing the SIDER task, and vice versa. This demonstrates the capability of our method to effectively inte-

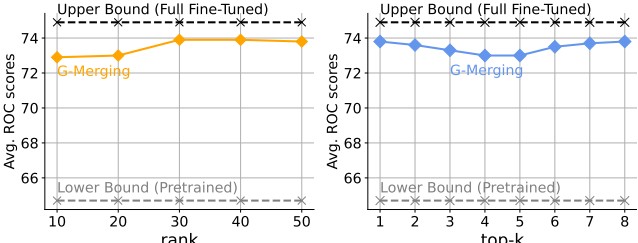

Figure 3: (a) Average scores over 8 tasks with varying ranks of the adapters. (b) Average scores over 8 tasks of top-k expert selection in our routing mechanism.

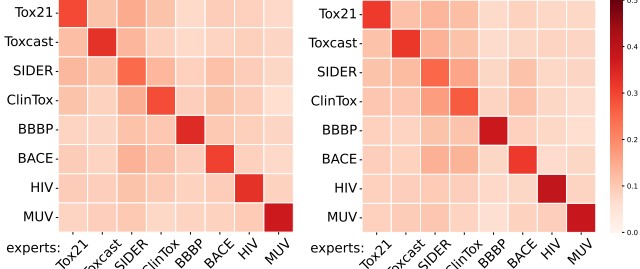

Figure 4: The MoE weight heatmaps illustrate the expert allocation patterns during inference on a target task, using two pre-trained models, GIN (contextpred) and GIN (edgepred).

grate knowledge across tasks. We further conduct experiments on the top-k expert selection strategy in our routing mechanism, which is a common design consideration in MoE models (see Appendix B for detailed settings). The overall results presented in Figure 3(b) (detailed in Appendix E Figure 7) show that as the number $K$ of selected experts increases from 1 to 8, the model performance first decreases and then increases. When relatively few task experts are selected, the source task cannot receive sufficient support from other tasks, and due to the heterogeneity of adapters, the effectiveness of the original adapter may also be affected, leading to performance degradation. In contrast, when more task experts are selected, the router can better fulfill its role by assigning appropriate balancing weights, allowing a large pool of similar task experts to collaboratively support the source task, thereby improving the performance. As a result, the above results suggest that G-Merging *possibly leverages shared knowledge* to promote generalization across tasks.

**Efficiency and Storage Analysis.** We examine how G-Merging can *reduce storage and computational cost*. As shown in Figure 3(a), our method remains effective under various settings of rank $r$, which is a hyperparameter related to the scale of adapters (see equation 6). It can be seen that, as the rank gradually increases, performance improves obviously when the rank is below 30, but decreases slightly when the rank exceeds 30. Empirically, our training time is approximately $1/8$ of finetuning a whole model (see Table 7 in Appendix E). Additionally, the number of parameters in Table 4 indicates that the scale of MoEAdapters is considerably smaller compared to a full GNN model.

Table 4: The parameter cost of the MoE adapters module.

| | The total number of parameters | |
|---|---|---|
| Rank | MoE adapters | Ratio(%) |
| 10 | 48000 | 2.58 |
| 20 | 96000 | 5.17 |
| 30 | 144000 | 7.75 |
| 40 | 192000 | 10.33 |
| 50 | 240000 | 12.92 |
| one full GNN model: 1857900 | | |

## 6 CONCLUSIONS

In this paper, we investigate the problem of graph model merging for the first time. We propose a method called G-Merging, which consolidates knowledge from task-specific fine-tuned models and achieves high performance while requiring lower storage and training costs. The effectiveness of G-Merging is validated by extensive experiments on 8 downstream datasets, which also indicate that graph model merging can be successfully achieved, like in models in the CV and NLP fields. Moreover, we demonstrate that fully utilizing the graph structure can significantly impact the performance of the merged model. Finally, we conclude that knowledge sharing and mutual benefit between tasks are feasible and promising for models on graph data. In the future, we will extend our work to more model merging and graph learning scenarios, such as graph continual learning.

## ACKNOWLEDGMENTS

The work of Ziyue Qiao was partially supported by the National Natural Science Foundation of China (No. 62406056) and the Guangdong Basic and Applied Basic Research Foundation (No. 2024A1515140114). The work of Qin Zhang was partially supported by the National Natural Science Foundation of China (No. 62576221) and the Guangdong Provincial Natural Science Foundation (No. 2025A1515010288).

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

# Appendix

In this appendix, we provide supplementary materials for this work. In Section A, we present detailed introduction of Topology-aware Wasserstein distance with related theoretical analysis. Sections B to G offer a comprehensive description of our experimental settings, including additional results and analyses. The limitations and furture works are in Appendix I

## A    TOPOLOGY-AWARE WASSERSTEIN DISTANCE

### A.1    BACKGROUND AND DEFINITION

Recall the definition of original Wasserstein Distance (Definition 1), we firstly discuss how to use the Wasserstein distance to measure the similarity between two sets of node embeddings in a graph. Given a graph $G(\mathcal{V}, \mathcal{E})$ with adjacency matrix $\mathbf{A}$ and two different node emmbedding matrix $\mathbf{X}^S$ and $\mathbf{X}^T$, corresponding to the two sets of node embeddings $\{\mathbf{x}_i^S\}_{i=1}^{|\mathcal{V}|}$ and $\{\mathbf{x}_i^T\}_{i=1}^{|\mathcal{V}|}$, we characterize these two sets using two unified distributions $\mu = \sum_{i=1}^{|\mathcal{V}|} \frac{1}{|\mathcal{V}|} \delta_{\mathbf{x}_i^S}$ and $\nu = \sum_{i=1}^{|\mathcal{V}|} \frac{1}{|\mathcal{V}|} \delta_{\mathbf{x}_i^T}$. , If we replace the distributions with $\mu$ and $\nu$ in Eq. 1, we obtain the **Wasserstein distance for node embeddings** as follows:

$$\mathcal{D}_{wdnode}(\mu, \nu) = \min_{\mathbf{T} \in \Pi(\mathbf{u}, \mathbf{v})} \sum_{i=1}^{|\mathcal{V}|} \sum_{j=1}^{|\mathcal{V}|} \mathbf{T}_{ij} \cdot c(\mathbf{x}_i^S, \mathbf{x}_j^T). \tag{10}$$

However, Eq. 10 depends solely on the node embeddings and not on the graph's topological structure. As discussed above, mass in $\mathbf{x}_i^S$ can be transported to any $\mathbf{x}_j^T \in \{\mathbf{x}_j^T\}_{j=1}^{|\mathcal{V}|}$ by transport plan $\mathbf{T}_{ij}$. Here, we intuitively assume that mass in $\mathbf{x}_i^S$ can only be transported to $\mathbf{x}_j^T$ if there is an edge between node $n_i$ and $n_j$ (i.e. $\mathbf{A}_{ij} = 1$). This constraint can be enforced by setting $\mathbf{T}_{ij} = 0$ whenever $\mathbf{A}_{ij} = 0$. In this way, we incorporate the graph's topological information into the WD, namely, the **Topology-aware Wasserstein Distance (TWD)**. The definition is given as follows:

**Definition 2** (Topology-aware Wasserstein distance). *Let $G(\mathcal{V}, \mathcal{E})$ be a graph with adjacency matrix $\mathbf{A}$ and two sets of node embeddings $\{\mathbf{x}_i^S\}_{i=1}^{|\mathcal{V}|}$ and $\{\mathbf{x}_i^T\}_{i=1}^{|\mathcal{V}|}$, represented by two unified distributions $\mu = \sum_{i=1}^{|\mathcal{V}|} \frac{1}{|\mathcal{V}|} \delta_{\mathbf{x}_i^S}$ and $\nu = \sum_{i=1}^{|\mathcal{V}|} \frac{1}{|\mathcal{V}|} \delta_{\mathbf{x}_i^T}$. The Topology-aware Wasserstein distance between $\mu$ and $\nu$ is then defined as:*

$$\mathcal{D}_{twd}(\mathbf{A}, \mu, \nu) = \min_{\mathbf{T} \in \Pi(A)} \sum_{i=1}^{|\mathcal{V}|} \sum_{j=1}^{|\mathcal{V}|} \mathbf{T}_{ij} \cdot c(\mathbf{x}_i^S, \mathbf{x}_j^T). \tag{11}$$

*where $\Pi(\mathbf{A}) = \{\mathbf{T} \in \mathbb{R}_+^{|\mathcal{V}| \times |\mathcal{V}|} \mid \mathbf{T}\mathbf{1}_{|\mathcal{V}|} = \frac{1}{|\mathcal{V}|} \cdot \mathbf{1}_{|\mathcal{V}|} \wedge \mathbf{T}^\top \mathbf{1}_{|\mathcal{V}|} = \frac{1}{|\mathcal{V}|} \cdot \mathbf{1}_{|\mathcal{V}|} \wedge \mathbf{T} \odot (\mathbf{1}_{|\mathcal{V}| \times |\mathcal{V}|} - \mathbf{A}) = \mathbf{0}_{|\mathcal{V}| \times |\mathcal{V}|}\}$ and $c(\cdot, \cdot)$ is the cost function. Compared to the original WD, the transport plan $\mathbf{T}$ is now constrained by graph adjacency matrix $\mathbf{A}$. Different matrix $\mathbf{A}$, node embedding $\mathbf{x}_i$, and cost function lead to different WD, and its obvious that $\mathcal{D}_{twd}(\mathbf{A}, \mu, \nu) > \mathcal{D}_{wd}(\mu, \nu)$.*

We define the matrix inner product $\langle \cdot, \cdot \rangle$ for $\mathbf{U}, \mathbf{V} \in \mathbb{R}^{m \times n}$ as $\langle \mathbf{U}, \mathbf{V} \rangle = \text{tr}(\mathbf{U}^\top \mathbf{V}) = \sum_{i,j} \mathbf{U}_{ij} \mathbf{V}_{ij}$. Furthermore, we introduce its more intuitive formulation, which serves as the alignment loss function in Section 4.2:

$$\text{TWD}(\mathbf{X}^S, \mathbf{X}^T, \mathbf{A}) = \mathcal{D}_{twd}(\mathbf{A}, \mu, \nu) = \min_{\mathbf{T} \in \Pi(A)} \sum_{i=1}^{|\mathcal{V}|} \sum_{j=1}^{|\mathcal{V}|} \mathbf{T}_{ij} \cdot c(\mathbf{x}_i^S, \mathbf{x}_j^T) = \min_{\mathbf{T} \in \Pi(A)} \langle \mathbf{T}, \mathbf{C} \rangle \tag{12}$$

where $\mathbf{C} \in \mathbb{R}^{|\mathcal{V}| \times |\mathcal{V}|}$ is the cost matrix with $\mathbf{C}_{ij} = c(\mathbf{x}_i^S, \mathbf{x}_j^T)$. A widely used option of the cost function is **cosine distance** $c(\mathbf{x}_i^S, \mathbf{x}_j^T) = \frac{1}{2}(1 - \cos(\mathbf{x}_i^S, \mathbf{x}_j^T))$ (Zhang et al., 2022; Xu et al., 2020; Chen et al., 2020), while others are not elaborated here. Additionally, we originally set $\mathbf{A}$ as the 1-hop adjacency matrix with self-loops, *i.e.*, $\mathbf{A}_{ij} = 1$ if and only if $i = j$ or there exists an edge between $i$ and $j$. The 1-hop adjacency matrix $\mathbf{A}$ is easy to extend to k-hop, like $\mathbf{A}^2$, $\mathbf{A}^3$, which may represent more global information of the graph structure. We conduct a supplementary experiment to examine this aspect in more detail, see Appendix G.

## A.2 Computation and Time Complexity

The computation of both WD and TWD requires solving for the optimal transport plan $\mathbf{P}$, a task that has been widely examined in the literature. In practice, an approximate solution to WD can be obtained in polynomial time by applying the Sinkhorn algorithm with an entropic regularization term.

Computing both WD and TWD involves solving for the optimal transport plan $\mathbf{P}$, a problem that has been extensively studied in prior work. In practice, an approximate solution to WD can be obtained in polynomial time by applying the Sinkhorn algorithm utlizing an entropic regularization term(Cuturi, 2013; Peyré et al., 2019; Dvurechensky et al., 2018).

For TWD, there exists an essential theoretical result for the iterative algorithm:

**Proposition 1.** *Let $\epsilon$ be a hypter-parameter, $\mathcal{H}(\cdot)$ be the entropy function, and assume that $0\log 0 = 0$. The solution to definition 2 with entropic regularization $\epsilon \cdot \mathcal{H}(\mathbf{A} \odot \mathbf{T})$ is unique and has the form*

$$\mathbf{T}_{ij} = \mathbf{u}_i \mathbf{A}_{ij} \mathbf{K}_{ij} \mathbf{v}_j \tag{13}$$

*where $\mathbf{K}_{ij} = \exp(-\mathbf{C}_{ij}/\epsilon)$ and $(\mathbf{u}, \mathbf{v}) \in \mathbb{R}_+^n \times \mathbb{R}_+^m$ are two unknown scaling variables.*

Based on this, the Sinkhorn algorithm is useful for computing the TWD by iteratively approximating the dual variables $\mathbf{u}$ and $\mathbf{v}$. However, the sparsity of $\mathbf{A}$ may lead to numerical instability or overflow during the iterative process. To mitigate this issue, we adopt an improved version of the Sinkhorn algorithm that performs computation in the log domain, which effectively alleviates such instability. The entire process involves a finite number of iterations. The complete algorithm, along with the theoretical justification of its correctness and feasibility, can be found in (Zhang et al., 2022).

We now analyze the computational complexity of the algorithm used to compute the TWD. Suppose that after certain iterations we get an approximate solution $\hat{\mathbf{T}}$ satisfying:

$$\langle \hat{\mathbf{T}}, \mathbf{C} \rangle \leq \mathrm{TWD}(\mathbf{X}^S, \mathbf{X}^T, \mathbf{A}) + \tau$$

According to (Altschuler et al., 2017), when $\tau = 4\epsilon \log(|\mathcal{V}|)$, the computational complexity of traditional Sinkhorn iterations to obtain $\hat{\mathbf{T}}$ for WD is:

$$O(n^2 |\mathbf{C}|_\infty^2 \tau^{-3}(\tau \log(s) + |\mathbf{C}|_\infty \log(n)))$$

where $|\mathbf{C}|_\infty := \max_{ij} \mathbf{C}_{ij}$ and $s := \sum_{ij} \mathbf{K}_{ij}$.

In TWD computation, $\mathbf{K}$ is replaced with $\mathbf{A} \odot \mathbf{K}$, so $s$ is replaced with:

$$s' := \sum_{ij} \mathbf{A}_{ij} \mathbf{K}_{ij} \leq s$$

Thus, the time complexity of the algorithm becomes:

$$O(n^2 |\mathbf{C}|_\infty^2 \tau^{-3}(\tau \log(s') + |\mathbf{C}|_\infty \log(n)))$$

This is smaller than the time complexity of original WD computation. In conclusion, the time complexity of our method is positively correlated with the number of nodes, the magnitude of the cost values, and the sparsity of the adjacency matrix. Therefore, the proposed method is theoretically efficient and feasible, especially on relatively small and sparse graph data (e.g., molecular graphs). As for the inference time, the TWD needs to be computed K times ( K is the number of tasks), with each computation differing only in the cost matrix (varying node features on the same graph). Therefore, the overall complexity is approximately linear with respect to K.

## B Experimental Setup Details

**Data Preprocessing and Splitting.** All input molecules in downstream task datasets are represented as SMILES strings (Weininger, 1988). We follow the preprocessing procedure described in Hu et al. (Hu et al., 2020), embedding the SMILE-formatted molecules into 120-dimensional node features and 3-dimensional edge features. We use scaffold (Bemis & Murcko, 1996), a splitting scheme based on the molecular graph structure, to divide the datasets into training, validation, and test sets following an 8:1:1 ratio. This scaffold split results in unseen structures in the validation and test sets, while common structures appear in the training set, which is closer to realistic model training and performance (Ramsundar et al., 2019).

**Training Settings and Computational Environment.** In our experiments, we employ two representative self-supervised pretraining strategies for graphs. One of them, Context Prediction, leverages subgraphs to infer their surrounding structures, with the goal of pretraining a GNN that places nodes occurring in similar structural contexts close to each other in the embedding space (Hu et al., 2020). Edge prediction leverages observed edge connections and node representations to predict whether a masked edge exists, encouraging the GNN to learn node embeddings that place connected nodes closer in the representation space (Hamilton et al., 2017). We run all our experiments on Nvidia 4090 GPUs equipped with 24GB RAM. We fine-tuned all parameters of the pre-trained models for 100 epochs, using binary cross-entropy loss. The learning rate was manually tuned from the set {0.01, 0.005, 0.001, 0.0005}, and the batch size was selected from {16, 32, 64}. During the merging stage, we train task-specific adapters for 30 epochs with the Adam optimizer. The learning rate was consistently set to 0.01, and the batch size selected from {16, 64, 256, 512}.

**Hyperparameter Strategies.** In the final test evaluation, the loss balancing parameter $\alpha$ in Eq. 6 is set to $\alpha = 1$. The scaling factor $\lambda$ in phase(I) of G-Merging (see Section 4.1) is searched to achieve optimal performance. The sensitivity of the aforementioned hyperparameters is further analyzed in Appendix F. For the TWD loss computation procedure, we set the hyperparameter $\epsilon = 0.1$, the threshold $\tau = 0.1$, and the maximum number of iterations to 100.

**Description of the Top-k Selection Experiments.** Top-k selection means that the MoE router sorts the assigned expert weights in descending order, keeps the top k weights, and sets all other weights to zero. Formally, let $\mathbf{w} = \{w_1, w_2, ..., w_K\}$ denotes the original MoE weights for K experts, the top-k selected weights are defined as :

$$\mathbf{w}_i^{\text{top-}k} = \begin{cases} w_i, & \text{if } w_i \text{ is among the top-}k \text{ largest weights in } \mathbf{w}, \\ 0, & \text{otherwise}, \end{cases} \quad i = 1, \dots, K$$

Then we compute the final MoE weights as $\mathbf{w}^{\text{final}} = \text{softmax}\{\mathbf{w}_i^{\text{top-}k}\}$ For this experiment, we use the GIN (contextpred) pretrained model, and keep all other setting same as in the main experiments.

**Other Details.** For the numerical stability of TWD computation, we normalize the elements of the cost matrix $\mathbf{C}_{ij} = \frac{1}{2}(1 - \cos(\mathbf{x}_i^S, \mathbf{x}_j^T))$ to the range $[0, 1]$ using max normalization. In most experiments, we set the adapter rank $r$ (Eq. 6) to 30, which provides a good trade-off between efficiency and performance.

## C  THOROUGH DESCRIPTION OF THE DATASETS

The downstream tasks are derived from the public MoleculeNet benchmark (Wu et al., 2018), a widely used collection for molecular machine learning. MoleculeNet comprises more than 700,000 compounds evaluated on diverse properties, which are mainly grouped into four categories: quantum mechanics, physical chemistry, biophysics, and physiology. For our experiments, we focus on eight datasets restricted to binary classification tasks within the biophysics and physiology domains. The statistical characteristics of these datasets are summarized in Table 5.

**BACE.** This dataset provides IC50 values and binary binding labels for 1,522 compounds targeting human $\beta$-secretase 1 (BACE-1) (Subramanian et al., 2016). Data are collected from published studies, with some molecules having crystal structures. Scaffold splitting is recommended for generalization.

**BBBP.** The BBBP dataset contains binary annotations for more than 2,000 chemical compounds, specifying whether they can penetrate the blood–brain barrier (Martins et al., 2012). It is commonly used in CNS drug development, with the scaffold split recommended.

**Tox21.** From the Tox21 initiative, this dataset offers qualitative toxicity labels for 8,014 compounds across 12 biological targets. It was featured in the 2014 Tox21 Data Challenge[1].

**ToxCast.** ToxCast contains high-throughput in vitro toxicity data for 8,615 compounds from over 600 bioassays (Richard et al., 2016), processed by MoleculeNet.

---

[1]Tox21 Challenge, https://tripod.nih.gov/tox21/challenge/.

Table 5: Statistics of the downstream graph datasets, including median, maximum, minimum, mean, and standard deviation of node numbers.

| dataset | Categories | Tasks | Molecules | median | max | min | mean | std |
|---------|-----------|-------|-----------|--------|-----|-----|------|-----|
| Tox21 | Physiology | 12 | 7831 | 14 | 114 | 1 | 16.5 | 9.5 |
| Toxcast | Physiology | 617 | 8575 | 14 | 103 | 2 | 16.7 | 9.7 |
| SIDER | Physiology | 27 | 1427 | 23 | 483 | 1 | 30.0 | 39.7 |
| ClinTox | Physiology | 1 | 1478 | 23 | 121 | 1 | 25.5 | 15.3 |
| BBBP | Physiology | 1 | 2039 | 22 | 63 | 2 | 22.5 | 8.1 |
| BACE | Biophysics | 1 | 1513 | 32 | 66 | 10 | 33.6 | 7.8 |
| MUV | Biophysics | 1 | 93087 | 24 | 44 | 6 | 24.0 | 5.0 |
| HIV | Biophysics | 17 | 41127 | 23 | 222 | 2 | 25.3 | 12.0 |

**SIDER.** SIDER catalogs side effects of 1,427 marketed drugs, grouped into 27 organ system classes based on MedDRA (Altae-Tran et al., 2017)[2].

**ClinTox.** ClinTox compares 1,491 compounds approved by the FDA vs. those that failed clinical trials due to toxicity (Artemov et al., 2016; Gayvert et al., 2016). It includes two tasks: predicting toxicity and FDA approval.

**MUV.** The MUV dataset consists of 90,000 compounds across 17 tasks, designed to reduce screening bias (Rohrer & Baumann, 2009). It serves as a benchmark for virtual screening models.

**HIV.** From the DTP AIDS Antiviral Screen, this dataset includes over 40,000 compounds tested for HIV inhibition[3]. "Active" and "moderately active" labels are merged; scaffold split is recommended.

## D    BASELINES DETAILS

Here we will elaborate on the baselines utilized in our main comparison experiment, as outlined in Table 1, 2, and 6.

**Full Fine-Tuned** refers to directly performing the downstream task using a model that has been fully fine-tuned on the task-specific dataset.

**Pretrained** means using the pre-trained model as a fixed graph representation encoder, equipped with a task-specific classification head taken from the corresponding fine-tuned model.

**Multi-task learning** aggregates datasets from multiple tasks into a unified training set and trains a single model to improve generalization and achieve better performance across tasks.

**Weight Average** (Choshen et al., 2022) straightforwardly averages the parameters of multiple fine-tuned models, which is computationally efficient but often results in inferior performance.

**Task Arithmetic** (Ilharco et al., 2022) first introduces the concept of *task vectors* and merges them into the pre-trained model with a hand-tuned scalar.

**Ties-Merging** (Yadav et al., 2023) Improves merging stability by resolving parameter conflicts based on tied weights across models.

**EMR-Merging** (Huang et al., 2025) is a training-free and high-performance model merging method, containing three steps: Elect, Mark, and Rescale.

**AdaMerging** (Yang et al., 2023) employs output entropy minimization to learn the merging coefficients without label supervision for each task vector (Task-wise AdaMerging) or for each layer (Layer-wise AdaMerging). AdaMerging++ is an enhanced version that applies Ties-Merging before learning the merging coefficients.

**Twin-Merging** (Lu et al.) decomposes the knowledge into shared and task-specific knowledge, where the exclusive knowledge can be compressed to enhance efficiency. Consequently, the router is trained to dynamically merge shared and task-specific knowledge based on the input. This approach significantly narrows the performance gap between pre-trained and fine-tuned models. However, it tends to perform suboptimally on graph-structured data.

---

[2]Medical Dictionary for Regulatory Activities, https://www.meddra.org/
[3]https://wiki.nci.nih.gov/spaces/NCIDTPdata/pages/158204006/AIDS+Antiviral+Screen+Data

## E  ADDITIONAL RESULTS

We provide additional experimental results that could not be included in the main text due to space constraints. Table 6 reports the outcomes of the primary comparison experiment based on the GCN (contextpred) pre-trained model. Figure 6 illustrates the detailed findings of the ablation study introduced in Section 5.2. Table 7 lists the complete results corresponding to Figure 3(a), supplemented with training time statistics. Figure 7 provides the detailed results corresponding to Figure 3(b). Figure 5 shows two additional heatmaps of the MoE weights corresponding to different pre-trained models beyond GIN (contextpred). Note that we compute the MoE weights for each input by summing the weights from the GraphAdapter and the layer-wise NodeAdapters.

Table 6: Test ROC-AUC score (%) of GCN models (contextpred) on downstream molecular property prediction tasks after merging fine-tuned models. ("$^*$" denotes performance surpassing that of the fine-tuned model.)

| Methods | Tox21 | Toxcast | SIDER | ClinTox | BBBP | BACE | HIV | MUV | Average |
|---|---|---|---|---|---|---|---|---|---|
| Full Fine-Tuned | 75.8 | 64.7 | 60.2 | 65.2 | 71.2 | 76.7 | 77.0 | 81.0 | 71.5 |
| Pretrained | 70.4 | 58.5 | 56.9 | 45.4 | 61.7 | 70.8 | 54.7 | 70.0 | 61.1 |
| Multi-Task Learning | 73.1 | 62.9 | 62.0 | 61.4 | 68.6 | 76.1 | 74.5 | 73.7 | 69.0 |
| Weight Average | 71.5 | 63.0 | 59.8 | 46.3 | 66.3 | 68.5 | 62.9 | 71.9 | 63.8 |
| Task Arithmetic | 71.7 | 63.0 | 59.9 | 47.1 | 66.2 | 69.2 | 62.4 | 72.1 | 63.9 |
| Ties-Merging | 70.4 | 58.7 | 57.9 | 42.6 | 61.1 | 72.3 | 57.8 | 72.7 | 61.7 |
| EMR-Merging | **73.8** | 61.3 | 60.8 | 53.2 | **70.3** | 73.2 | 72.7 | 66.1 | 66.4 |
| AdaMerging | 68.4 | 59.1 | 56.3 | 34.4 | 61.2 | 63.7 | 61.6 | 65.1 | 58.7 |
| Twin-Merging | 71.4 | 59.8 | 58.5 | 53.3 | 62.4 | 57.4 | 57.4 | 72.0 | 61.5 |
| **G-Merging-s (Ours)** | 73.0$_{\pm0.2}$ | 63.1$_{\pm0.1}$ | 61.9$^*_{\pm0.3}$ | 62.2$_{\pm2.0}$ | 69.8$_{\pm0.2}$ | **73.4**$_{\pm0.3}$ | 68.5$_{\pm1.5}$ | **78.8**$_{\pm0.5}$ | 68.8 |
| **G-Merging (Ours)** | 73.0$_{\pm0.2}$ | **63.2**$_{\pm0.1}$ | **62.0**$^*_{\pm0.3}$ | **62.6**$_{\pm4.2}$ | 69.8$_{\pm0.2}$ | 73.3$_{\pm0.6}$ | **68.6**$_{\pm1.5}$ | **78.8**$_{\pm0.5}$ | **68.9** |

Table 7: Efficiency and Storage Analysis of GIN (contextpred) models.

| Methods | Tox21 | Toxcast | SIDER | ClinTox | BBBP | BACE | HIV | MUV | Average | times |
|---|---|---|---|---|---|---|---|---|---|---|
| Full Fine-Tuned | 78.0 | 64.8 | 62.5 | 74.0 | 69.6 | 86.8 | 79.6 | 83.9 | 74.9 | about 400+ min |
| Multi-Task Learning | 75.5 | 63.4 | 62.8 | 64.9 | 66.4 | 84.7 | 74.8 | 77.5 | 71.2 | 144 min 45 s |
| Pretrained | 68.9 | 63.3 | 58.1 | 61.9 | 55.3 | 78.5 | 59.1 | 72.3 | 64.7 | 0 min |
| G-Merging(r=10) | 76.0 | 65.6 | 63.1 | 78.9 | 66.0 | 83.8 | 73.3 | 75.9 | 72.9 | 59 min 4 s (single 4090 GPU) |
| G-Merging(r=20) | 77.0 | 65.8 | 64.2 | 71.7 | 66.7 | 85.6 | 73.3 | 78.8 | 73.0 | 58 min 28 s (single 4090 GPU) |
| G-Merging(r=30) | 77.4 | 65.8 | 64.4 | 74.1 | 67.1 | 86.9 | 74.5 | 81.2 | 73.9 | 57 min 56 s (single 4090 GPU) |
| G-Merging(r=40) | 77.5 | 65.8 | 64.4 | 72.9 | 67.5 | 87.2 | 74.1 | 81.7 | 73.9 | 58 min 49 s (single 4090 GPU) |
| G-Merging(r=50) | 77.5 | 65.8 | 64.3 | 71.2 | 67.6 | 86.6 | 75.4 | 82.7 | 73.8 | 58 min 10 s (single 4090 GPU) |

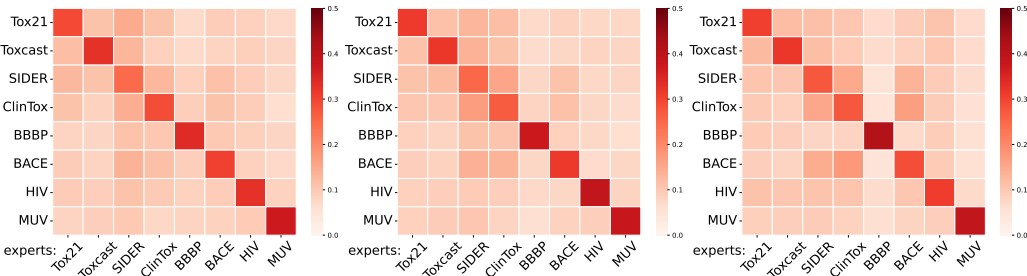

Figure 5: MoE Weights Heatmap. The heatmap illustrates the weights assigned by the routing mechanism of G-Merging between different testing tasks (rows) and task-specific expert adapters in the MoE framework (columns). Each value represents the weight distribution, indicating the contribution of each expert to a given task. Darker colors correspond to higher weights, highlighting more influential expert-task relationships. From left to right, the three figures correspond to three pre-trained model settings: GIN (ContextPred), GIN (EdgePred), and GCN (ContextPred), respectively.

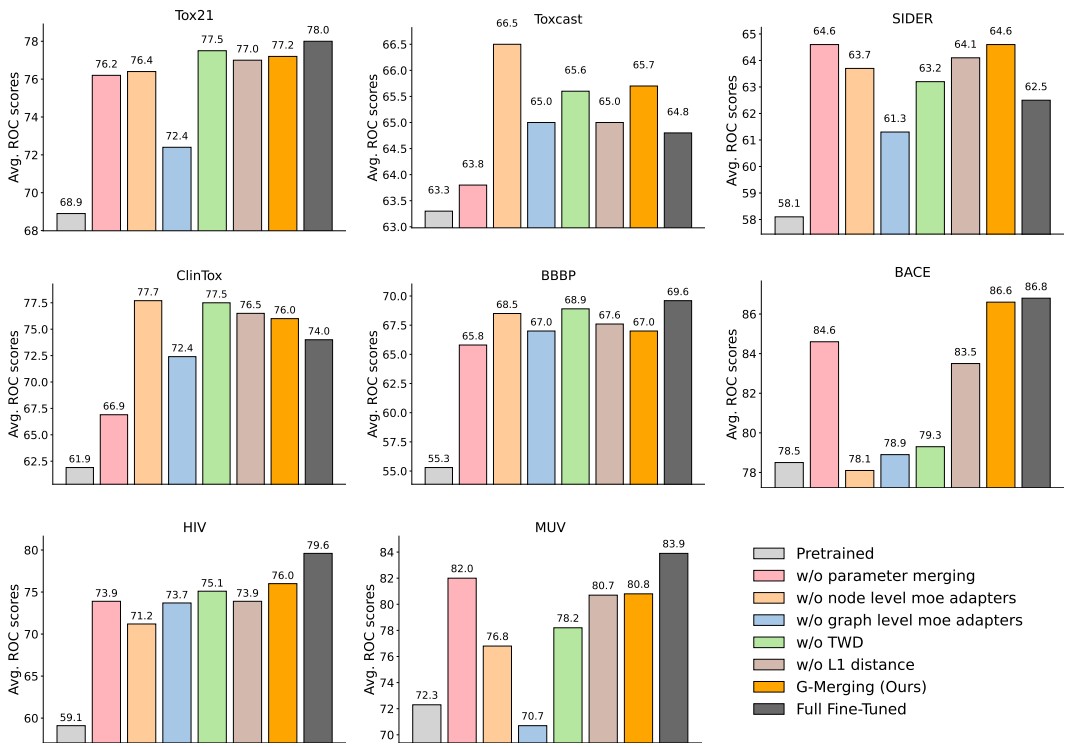

Figure 6: The detailed results of ablation studies in Section 5.2. We evaluate five ablated variants of our method, each with a specific component removed, across 8 downstream tasks.

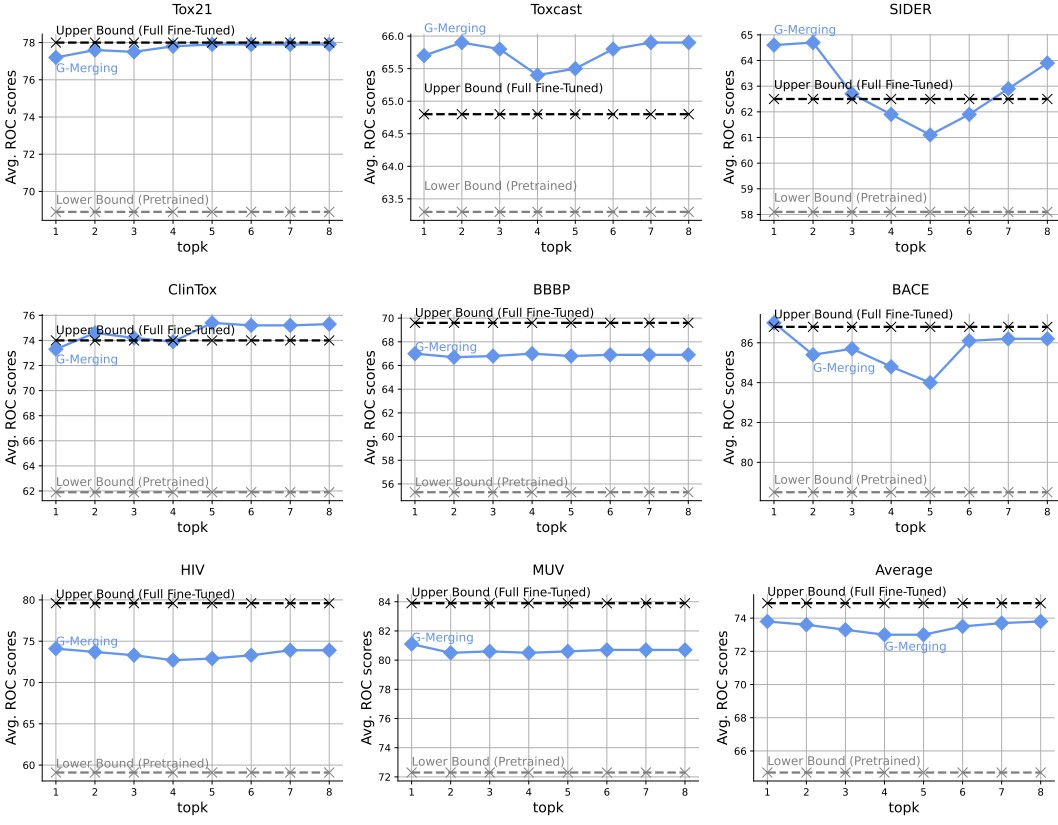

Figure 7: The impact of Top-k selection in the MoE structure on the performance of G-Merging, tested on 8 tasks and their average.

# F HYPERPARAMETER SENSITIVITY ANALYSIS IN G-MERGING

We further conduct experiments about two inevitable hyperparameters in G-Merging: task vectors scalar $\lambda$ and loss balance $\alpha$. As shown in Figure 8 and 9, We present the performance variety of G-Merging with $\lambda$ ranging from $0.05$ to $0.25$ and $\alpha$ ranging from $10^{-6}$ to $10$.

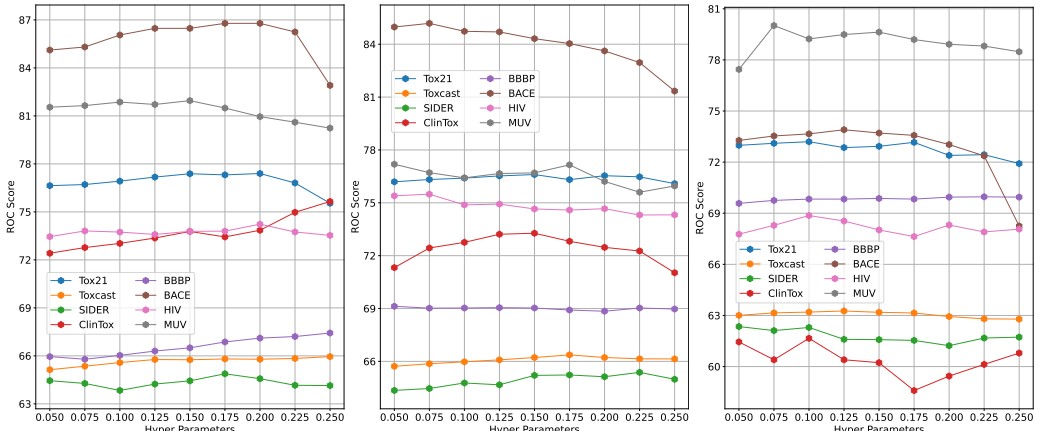

Figure 8: Effect of hyperparameter $\lambda$. Lines of different colors represent the performance on each task. From left to right, the three figures correspond to three pre- trained model settings: GIN (ContextPred), GIN (EdgePred), and GCN (ContextPred), respectively

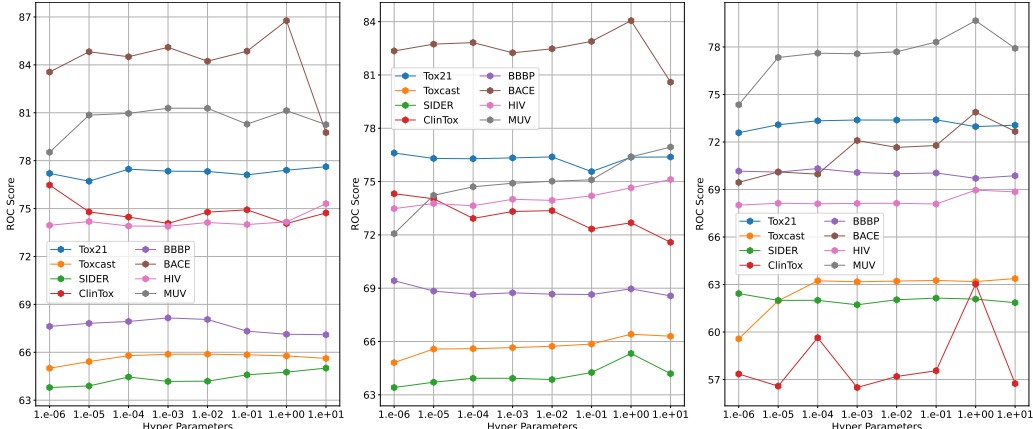

Figure 9: Effect of hyperparameter $\alpha$. Lines of different colors represent the performance on each task. From left to right, the three figures correspond to three pre- trained model settings: GIN (ContextPred), GIN (EdgePred), and GCN (ContextPred), respectively

# G VARIOUS ADJACENCY MATRICES IN CALCULATING TWD

Recalling Eq. 5, the topology information incorporated into TWD is based on the standard 1-hop adjacency matrix $\mathbf{A}$. However, this matrix can be easily extended to a $k$-hop adjacency matrix, which may encode different aspects of the topological structure. To this end, we investigate the impact of using different adjacency matrices on the performance of TWD. As shown in Figure 10, the 1-hop adjacency matrix generally performs better overall, while other adjacency matrices achieve competitive performance on certain specific tasks.

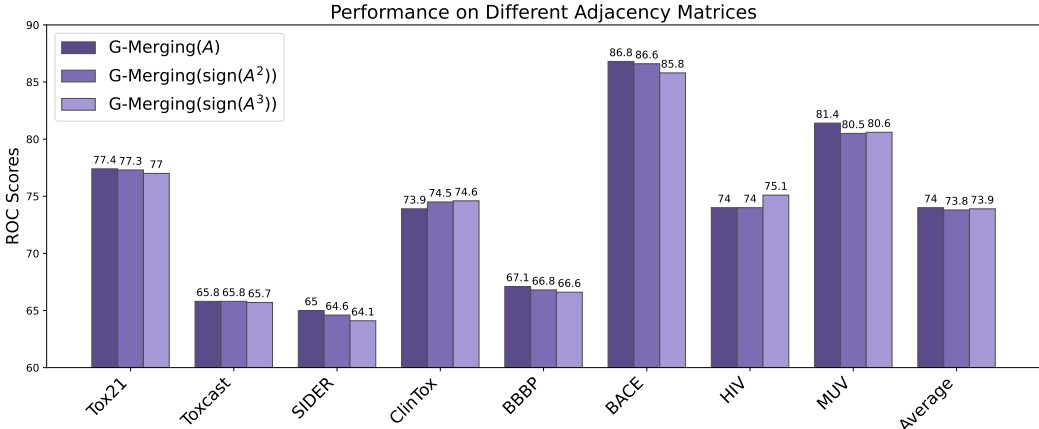

Figure 10: Investigating the Impact of Adjacency Matrices on TWD. The GIN (contextpred) pretrained model is used in this experiment.

## H  G-MERGING ON HETEROGENEOUS AND NON-MOLECULAR GRAPH DOMAINS.

To further evaluate the generalization of G-Merging beyond molecular property prediction, we conduct experiments on nine heterogeneous downstream tasks, covering transportation networks, academic graphs, movie actor networks, social networks, and Reddit interaction graphs. These tasks include both node-level and graph-level classification and therefore provide a comprehensive assessment of performance outside the molecular domain. First, we presents the experimental setup here.

**Pretraining.** Following the protocol of GCC (Qiu et al., 2020), we adopt a publicly available pretrained GNN model trained jointly on a mixture of heterogeneous graph datasets, including: Academic networks (Academia, DBLP-SNAP, DBLP-NetRep), Entertainment networks (IMDB), and Social networks (Facebook, LiveJournal). These datasets differ substantially in structure and semantics, providing a suitable basis for evaluating the generalizability of our method beyond molecular domains.

**Downstream Datasets.** We fine-tune the pretrained model on nine downstream datasets covering both node-level and graph-level prediction tasks. For node-level classification, we use four datasets: USA Airport, Europe Airport, Brazil Airport, three Airline transportation networks (over 1k nodes), where labels correspond to airport activity levels (4 classes); and H-index, co-authorship subgraph from Open Academic Graph, where labels indicate whether an author's h-index is above or below the median. For graph-level classification, we use five datasets: IMDB-Binary and IMDB-Multi, movie actor collaboration graphs (small, dense, 2–3 classes); COLLAB, medium-sized scientific collaboration networks; Reddit-Binary and Reddit-5K, Large-scale sparse Reddit discussion graphs with diverse interaction patterns. These datasets were also used in GCC (Qiu et al., 2020) as downstream tasks to evaluate the generalization of GNN models pretrained using their proposed strategy. And we believe that this setting already spans both node-level and graph-level tasks over diverse and heterogeneous graph topologies.

**Merging Procedure.** For each dataset, we fine-tune one model from the same pretrained checkpoint. Then we apply G-Merging and two baseline methods (Weight Average and Task Arithmeti) to merge nine finetuned models. All merged models are evaluated on their corresponding tasks using F1-score. The results are presented below:

Table 8: Performance comparison on diverse non-molecular graph benchmarks.

| Method | usa_airport | europe_airport | brazil_airport | h-index | imdb-binary | imdb-multi | collab | rdt-b | rdt-5k |
|---|---|---|---|---|---|---|---|---|---|
| Pre-trained | 41.68 | 37.85 | 56.54 | 56.84 | 59.60 | 39.27 | 61.44 | 59.70 | 32.07 |
| Weight Average | 56.49 | 56.32 | 45.09 | 65.29 | 60.40 | 39.88 | 62.12 | 61.32 | 33.09 |
| Task Arithmetic | 58.20 | **58.31** | 42.90 | 69.10 | 59.80 | 42.60 | 60.80 | 67.80 | 27.80 |
| **G-Merging (Ours)** | **59.01** | 57.24 | **60.52** | 72.87 | **61.29** | **47.30** | **62.22** | **70.57** | **39.11** |

We observe that G-Merging achieves the best performance on most of datasets and improves substantially over baselines on both node-level and graph-level tasks. These results demonstrate that G-Merging generalizes well to non-molecular graph domains such as transportation, academic, and social networks, and it remains effective on diverse task types, including large sparse graphs and multi-class graph classification.

Overall, this study confirms that G-Merging is not restricted to molecular property prediction. Our method successfully merges models fine-tuned on a broad variety of graph domains and task settings, which confirming its applicability under heterogeneous and non-molecular scenarios.

## I    LIMITATIONS AND FUTURE WORKS

Our proposed G-Merging framework demonstrates promising results in merging fine-tuned graph models and transferring knowledge across tasks. However, our setting assumes the task-specific models used for merging are all fine-tuned from a shared pretrained GNN model. A more general scenario, where these models may originate from different checkpoints or even differ in architecture, remains an open direction for future exploration. Also, the use of a fixed number of fine-tuned models may limit adaptability to evolving or unseen tasks.

In future work, we plan to study a more generalized graph model merging framework by developing adapter alignment strategies that can reconcile representations from different backbones, and also explore unified routing mechanisms that remain effective across diverse model types. Furthermore, we plan to extend the framework with continual learning capabilities to enable dynamic adapter composition, improving the model's practicality in real-world scenarios.

