# OpenReview forum: "G-Merging: Graph Models Merging for Parameter-Efficient Multi-Task Knowledge Consolidation"
_ICLR.cc/2026/Conference — ICLR 2026 Poster_

### Official Review · Reviewer_VpK5 · 2025-10-21

**Soundness:** 2
**Presentation:** 3
**Contribution:** 3
**Rating:** 6
**Confidence:** 4

**Summary:**

This paper introduces G-Merging, a novel framework for consolidating multiple task-specific fine-tuned Graph Neural Network (GNN) models into a unified multi-task model with parameter efficiency. The proposed approach consists of three key components: (1) Task Arithmetic for coarse parameter merging, (2) Topology-aware Wasserstein Distance (TWD) for aligning embeddings and training lightweight task-specific adapters, and (3) a training-free MoE router that dynamically selects task-specific adapters based on structural similarity. The authors position this work as the first systematic approach to address the model merging problem specifically for graph neural networks.

**Strengths:**

1. This is the first work to systematically address the model merging problem specifically for graph neural networks. The proposed three-stage framework (Task Arithmetic, TWD-based alignment, MoE router) is logically sound with each component complementing the others.
2. By only fine-tuning a small number of adapter parameters rather than the entire model, G-Merging significantly reduces computational and storage costs for multi-task deployment, aligning with current trends in parameter-efficient fine-tuning.
3. The evaluation across multiple molecular property prediction tasks demonstrates G-Merging's effectiveness in consolidating knowledge from multiple task-specific models. Ablation studies validate the contribution of each component.
4. The paper is well-structured with clear methodology descriptions and intuitive visualizations that make complex technical content accessible.

**Weaknesses:**

1. The current approach assumes all tasks share the same output dimension, preventing it from handling tasks with different output dimensions (e.g., binary vs. multi-class classification). This represents a significant limitation since real-world applications often involve diverse task types with varying output structures.
2. The method involves multiple hyperparameters (task vector scaling factor, loss balancing weights, adapter rank, etc.). While the paper provides some sensitivity analysis, there's insufficient guidance on how to select these hyperparameters based on task characteristics.
3. Current experiments focus exclusively on molecular property prediction tasks (all graph-level binary classification), lacking validation on other graph task types (e.g., node classification, link prediction).
4. The paper primarily compares against model merging approaches but lacks comprehensive comparison with parameter-efficient fine-tuning (PEFT) methods for multi-task learning (e.g., LoRA-based approaches).
5. While the topology-aware Wasserstein distance is proposed, there's minimal theoretical analysis of its properties, relationship to graph structural characteristics, or convergence guarantees. Additional theoretical insights would strengthen the methodological contribution.

**Questions:**

1. The paper assumes all tasks use the same pre-trained GNN backbone. How would G-Merging handle cases where different tasks employ different GNN architectures? If not currently supported, what modifications would be needed to extend the framework?
2. In the MoE router, how is "structural similarity" formally defined and computed? Is there theoretical justification for this similarity metric's effectiveness? Were different similarity measures experimentally compared?
3. How does G-Merging perform when merging models trained on graph datasets with significantly different structural distributions (e.g., social networks vs. molecular graphs)? Are there specific mechanisms to handle such task heterogeneity?
4. The paper mentions that TWD leverages graph topological information, but the ablation study only shows results with different adjacency matrix powers. Could you provide more detailed analysis on how different topological aspects (e.g., degree distribution, community structure) affect the merging performance?

If the authors provide satisfactory responses to these concerns during the rebuttal phase, I would be prepared to revise my recommendation to a stronger accept.

---

> ### Author Response · Authors · 2025-11-20
> **Response to Weakness 1 & 2 & 3 & 4**
>
> Thank you for the valuable and constructive feedback! We provide our detailed responses as follows.
>
> **W1**
>
> We want to claim that, in our model merging setting, all downstream tasks have different output dimensions.
> Every task is equipped with a task-specific output head that is kept fixed during the entire model merging procedure.
> The merging is performed only on the graph encoder part of the finetuned models.
> Please refer to our problem definition in Section 3, Line 145,146.
>
> >Then, $f_{θ_m}$ as a shared graph encoder, combined with the task-specific classification heads from the fine-tuned models, is used to perform multi-task inference.
>
> This is also illustrated in our framework in Figure 2.
> Specifically, the output shape of each task is listed in Table 5, where the number of binary classification tasks corresponds to the output dimensionality of each finetuned model. In other words, each downstream task consists of a different number of subtasks, where each subtask is a binary graph classification.
>
> In addition, it is clearly feasible that downstream tasks involve multi-class classification. For instance, in our additional experiments (highlighted in blue, added in Appendix H of the revised paper), the Airport dataset is a 4-class node classification task, while the H-index dataset is a 2-class node classification task, resulting in different output dimensions.
>
>
> **W2**
>
> According to the sensitivity analysis results in Appendix F, the performance is not particularly sensitive to the task vector scaling factor $\lambda$ and the loss balance coefficient $\alpha$.
> The performance curves appear overall smooth, indicating stability with respect to these hyperparameters.
> In addition, the loss balance $\alpha$ used in our main experiments is simply set to 1, a standard and intuitive choice.
> We did not deliberately tune it for performance gains.
> As for the task vector scaling factor $\lambda$, manually tuning it for optimal performance is not an essential component of the original Task Arithmetic method.
> And we simply adopt the same setting in their paper.
>
> Moreover, the adapter rank $r$ represents the hidden dimension of each adapter layer and directly determines the total number of additional parameters.
> It is intuitive that a higher $r$ leads to stronger modeling capacity and worse storage and inference time.
> As shown in Table 7, we report the model performance under various rank settings.
> Noted that reducing storage and computation while maintaining downstream performance is one of the core goals of the model merging problem.
> We aim for a better tradeoff between performance and efficiency and set the $r$=30 in our main experiments, the middle value in Table 7.
> Therefore, rank $r$ is not a hyperparameter tuned specifically for performance, but rather a practical choice based on tradeoff considerations.
>
>
> **W3**
>
> Thanks for the suggestion. We have further conducted experiments on mixed task types, including both node-level and graph-level prediction. We have added the details of this experiment in Appendix H of the revised paper (highlighted in blue), and we briefly summarize it here.
> Specifically, we evaluate our method on nine downstream tasks spanning transportation networks, academic graphs, movie actor networks, social networks, and Reddit interaction graphs. Among them, four are node-level classification tasks and five are graph-level classification tasks. Then we merge the nine fine-tuned models using G-Merging and baseline methods. We report the F1-score on each task as shown below.
>
> | Method              | usa_airport | europe_airport | brazil_airport | h-index   | imdb-binary | imdb-multi | collab    | rdt-b     | rdt-5k    |
> | ------------------- | ------------ | ----- | ------------ | --------- | ----------- | ---------- | ------ | --------- | --------- |
> | **Pre-trained**     | 41.68        | 37.85           | 56.54           | 56.84     | 59.60       | 39.27      | 61.44     | 59.70     | 32.07     |
> | **Weight Average**  | 56.49        | 56.32           | 45.09           | 65.29     | 60.40       | 39.88      | 62.12     | 61.32     | 33.09     |
> | **Task Arithmetic**  | 58.20        | **58.31**       | 42.90           | 69.10     | 59.80       | 42.60      | 60.80     | 67.80     | 27.80     |
> | **G-Merging**       | **59.01**    | 57.24           | **60.52**       | **72.87** | **61.29**   | **47.30**  | **62.22** | **70.57** | **39.11** |
>
> The results show that G-Merging outperforms all baseline methods, demonstrating that our approach remains effective across diverse graph task types.
>
> **W4**
>
> We want to claim that, our model merging setting requires a set of finetuned models and no access to labels for downstream tasks, which differs significantly to PEFT. PEFT aim to efficiently inprove the peformance to specific task using training on task labels, while our method aim to obtain a multi-task model. Thus, PEFT method isnot appropriate as baseline for comparison.

---

> ### Author Response · Authors · 2025-11-20
> **Response to Weakness 5, Question 1 & 2**
>
> **W5**
>
> Thanks for this insightful comment. While a full theoretical analysis is beyond the scope of the current work, we provide empirical evidence that TWD effectively captures meaningful structural information relevant for model merging.
>
> Specifically, in our previous analysis of the MoE routing weights (Figure 4 in the paper), we observed that the TWD-based routing strongly correlates with the task graph structural complexity. We constructed a structural similarity matrix computed from statistics of graph node counts (median, max, min, mean, std) (see Table 5 in Appendix). Then we computed the Spearman and Pearson correlation coefficients between the TWD-based routing weights matrix and the structural similarity matrix. The results are shown below:
>
> | Spearman Correlation | Pearson Correlation | t-value  | p-value       |
> |----------------------|----------------------|----------|----------------|
> | 0.4264             | 0.5368               | 5.0101   | 4.8100e-06     |
>
> The Spearman and Pearson correlations between the TWD-based MoE routing weights and task graph structural characteristics were positive and statistically significant (p < 0.001), indicating that TWD effectively reflects both task characteristics and graph structural properties.
>
> These results suggest that, although formal theoretical guarantees have not yet been derived, TWD inherently leverages graph topology through constrainted optimal transport. This further supports the practical effectiveness of TWD in integrating structural information for model merging. We agree that further theoretical characterization is a valuable direction for future work.
>
>
>
> **Q1**
>
> To the best of our knowledge, almost all existing model merging research (in the CV and NLP fields) assumes that the models share the same architecture. These works typically perform model merging under the pretrain-finetune paradigm, i.e., each model to be merged is fine-tuned from the same pretrained checkpoint and therefore naturally has the same architecture.
> That said, if one wishes to merge models with different architectures, a feasible approach could be to first distill all models into a unified architecture and then perform conventional model merging. Overall, we consider this an interesting direction for future work.
>
> **Q2**
>
> In our MoE router method, we measure structural similarity by computing the TWD and then taking its inverse; please refer to Equation 8 in the paper. Theoretically, if two tasks are similar, the node embeddings processed by their respective experts should be similar within their neighborhoods, leading to smaller TWD values.
>
> Furthermore, we conduct experiments comparing the similarity measured by TWD and two baselines (directly using cosine similarity, using WD as similarity measurer). The results are shown below, using GIN(contextpred) pretrained model and evaluated by F1-score:
>
> | method | tox21   | toxcast | sider   | clintox | bbbp    | bace    | hiv     | muv     |
> | --------------- | ------- | ------- | ------- | ------- | ------- | ------- | ------- | ------- |
> | **similarity based on TWD**               | 77.3826 | 65.7579 | 64.3668 | 74.1025 | 67.0718 | 86.5241 | 74.0936 | 81.3726 |
> | **similarity based on WD**    |  76.6997 | 65.1437 | 62.6778 | 74.4142 | 65.6539 | 81.6032 | 74.1059 | 79.5371 |
> | **consine similarity**              | 74.8946 | 63.1567 | 61.7329 | 74.0258 | 65.9876 | 85.5697 | 73.5896 | 75.6598 |
>
> We observe that replacing TWD with cosine similarity or WD leads to a clear drop in performance, demonstrating the effectiveness of our TWD-based structural similarity measure.

---

> ### Author Response · Authors · 2025-11-20
> **Response to Question 3 & 4**
>
> **Q3**
>
> Thanks for your suggestion. As in previous work on computer vision and natural language processing [1,2], model merging is typically performed under the pretrain-finetune paradigm.
> Molecular property prediction is the usually widely adopted downstream task in the pretrain-finetune setting for GNNs and graph data [3,4]. Notably, many existing studies on graph finetuning methods also focus on molecular datasets, thus we choose the dataset same as [3,4] in this paper.
>
> We have now further conducted experiments on nine downstream tasks with more diverse graph domains and task heterogeneity, following the pretrain-finetune paradigm in GCC [5]. These nine datasets span domains such as transportation networks, academic collaboration graphs, movie and entertainment networks, and social network graphs. Although we were unable to find existing benchmarks that jointly include both social network data and molecular graphs as you suggested, we believe that this setting already provides a sufficiently diverse and heterogeneous collection of graph topologies to robustly evaluate our method.
>
> Specifically, we evaluate our method on nine downstream tasks, including four node-level classification tasks and five graph-level classification tasks. Then we merge the nine fine-tuned models using G-Merging and baseline methods. We report the F1-score on each task as shown below.
>
> | Method              | usa_airport | europe_airport | brazil_airport | h-index   | imdb-binary | imdb-multi | collab    | rdt-b     | rdt-5k    |
> | ------------------- | ------------ | ----- | ------------ | --------- | ----------- | ---------- | ------ | --------- | --------- |
> | **Pre-trained**     | 41.68        | 37.85           | 56.54           | 56.84     | 59.60       | 39.27      | 61.44     | 59.70     | 32.07     |
> | **Weight Average**  | 56.49        | 56.32           | 45.09           | 65.29     | 60.40       | 39.88      | 62.12     | 61.32     | 33.09     |
> | **Task Arithmetic**  | 58.20        | **58.31**       | 42.90           | 69.10     | 59.80       | 42.60      | 60.80     | 67.80     | 27.80     |
> | **G-Merging**       | **59.01**    | 57.24           | **60.52**       | **72.87** | **61.29**   | **47.30**  | **62.22** | **70.57** | **39.11** |
>
> The results demonstrate that our method remains effective across diverse graph domains and topological heterogeneity. We have added the detailed description and analysis of this experiment in Appendix H of the revised paper (highlighted in blue).
>
> [1] Prateek Yadav, Derek Tam, Leshem Choshen, Colin A Raffel, and Mohit Bansal. Ties-merging: Resolving interference when merging models. Advances in Neural Information Processing Systems, 36:7093–7115, 2023.
> [2] Enneng Yang, Zhenyi Wang, Li Shen, Shiwei Liu, Guibing Guo, Xingwei Wang, and Dacheng Tao. Adamerging: Adaptive model merging for multi-task learning. arXiv preprint arXiv:2310.02575, 2023.
> [3] WANG Zhili, DI Shimin, CHEN Lei, and ZHOU Xiaofang. Search to fine-tune pre-trained graph neural networks for graph-level tasks. In 2024 IEEE 40th International Conference on Data Engineering (ICDE), pages 2805–2819. IEEE, 2024.
> [4] Hu, Weihua, et al. Strategies for pre-training graph neural networks. arXiv preprint arXiv:1905.12265 (2019).
> [5] Jiezhong Qiu, Qibin Chen, Yuxiao Dong, Jing Zhang, Hongxia Yang, Ming Ding, Kuansan Wang, and Jie Tang. Gcc: Graph contrastive coding for graph neural network pre-training. In Proceedings of the 26th ACM SIGKDD international conference on knowledge discovery & data mining, pages 1150–1160, 2020.
>
> **Q4**
>
> We thank the reviewer for this insightful question. While it is indeed interesting to explore the influence of specific topological properties such as degree distribution or community structure on merging performance, due to space and time constraints we have not conducted these detailed analyses in the current revision.
>
> However, the theoretical design of TWD inherently incorporates topological information through neighborhood-aware node embeddings and graph-constrained optimal transport. We also believe that the results of the ablation studies with various adjacency matrix powers demonstrate that TWD captures both local structural similarity and broader neighborhood topology, suggesting that topological properties like degree distribution and community structure are likely being leveraged.
>
> We agree with that a more fine-grained analysis on topological aspects would be valuable, and we consider it a promising direction for future work.

---

> > ### Comment · Reviewer_VpK5 · 2025-11-23
> >
> > Thank you to the author for the detailed response, which addressed my concerns. I look forward to your future works.

---

> > > ### Author Response · Authors · 2025-11-24
> > >
> > > Thank you for acknowledging our work and for taking the time to review it. We greatly appreciate your feedback and are glad that our response addressed your concerns.

---

### Official Review · Reviewer_tomN · 2025-10-27

**Soundness:** 3
**Presentation:** 2
**Contribution:** 2
**Rating:** 6
**Confidence:** 4

**Summary:**

The paper introduces a strategy for model merging in graph neural networks (GNNs). In particular, the considered setting includes a pre-trained GNN, trained on some general task, which then gets adapted into multiple specialized versions through fine-tuning for specific tasks. The goal is then to merge the fine-tuned versions together to obtain a single model that can perform all tasks with comparable performance. The proposed method involves first merging parameters with task vectors, then adding adapter modules in between graph convolutions (and training them), and finally a training-free mixture of expert approach to mix the contributions from different adapters at inference time. A novel loss based on Wasserstein distance is used to train the adapters with the goal of reducing the difference in the distribution of the representations between the unified and fine-tuned models.
The experiments consider 8 binary graph classification datasets regarding molecule property prediction, and compare against existing model averaging techniques (Weight Averaging, Task Arithmetic, Ties-Merging, and EMR-Merging, AdaMerging, Twin-merging). Additional baselines are given by multi-tasks learning, individual fine-tuning, and the pre-trained model. Results show that the merged model has similar performance to the individual fine-tuned ones.
An ablation study shows the importance of all components in the proposed method.

**Strengths:**

- Extensive experimental section with many baselines.
- Informative ablation experiments to showcase the inner workings of the method (router heatmaps and top-k expert selection plots)

**Weaknesses:**

- The graphs considered in the experimental evaluation mostly have a very small number of nodes.
- The proposed Wasserstein loss can be computationally prohibitive for large graphs.

**Questions:**

-  Oversmoothing is a well known problem of GNNs. The Wasserstein loss introduced in this paper seems to actually encourage oversmoothing (it would be minimized if all representations were the same). Could the authors please comment on this?
- The proposed Wasserstein loss has a complexity which is quadratic in the number of nodes (as shown in the appendix). How does this behave for large graphs? Would it be possible to add some runtime values for some of the experiments?
- One of the properties of the proposed Wasserstein loss is that it is "aware of graph topology" (i.e., constrained by the graph adjacency matrix). Have the authors tried removing this constraint and seeing how performance would change? I agree that this seems like a principled choice, but at the same time giving more freedom to the transport plan may provide benefits?
- Could the authors expand on the practicality of the proposed method? It requires a pre-trained model, and several fine-tuned versions of it. While this is a somehow realistic scenario for large language models, is this something that happens when deploying GNNs in real-world applications?
- If possible, as the experiments have been run over 5 random seeds, could you add numbers for the variance/standard deviation? I think this is an important factor to validate the effectiveness of the method.

---

> ### Author Response · Authors · 2025-11-20
> **Response to Question 1 & 2 & 3**
>
> Thank you for the valuable and constructive feedback. Below we provide our responses.
>
> **Q1**
>
> Thank you for raising this important point. While it is true that minimizing an unconstrained Wasserstein distance may admit a degenerate solution where all node representations collapse to the same vector (i.e. oversmoothing), this concern does not apply to our method due to two reasons as follow:
>
> (1) **TWD itself does not encourage global smoothing due to topology constraints.** Our TWD (refer to Equation 5) enforces:
>
> $$
> \mathbf{T} \odot (\mathbf{1}_ {|\mathcal{V}| \times |\mathcal{V}|} - \mathbf{A}) = \mathbf{0}_ {|\mathcal{V}| \times |\mathcal{V}|}
> $$
>
> i.e., mass can only be transported along edges, which make it differ to original WD. This means: i) TWD aligns local neighborhoods, not all nodes globally. ii) On sparse graphs (e.g., molecular graphs), enforcing consistent collapse across all neighborhoods becomes extremely difficult. Thus TWD optimizes local structural consistency, not global homogeneity.
>
> (2) **The low-rank adapter architecture makes collapse architecturally infeasible.** The adapter (refer to equation 3 and 4) is a low-rank ReLU MLP added subtractively after each GNN layer:
>
> $$
> \mathbf{H}^{(l)} = \mathbf{H}_ {\text{uni}}^{(l)} - f_{\text{adap}}\left( \mathbf{H}_ {\text{uni}}^{(l)} \right).
> $$
>
> Note that only the lightweight adapter is trainable, while the majority of parameters (i.e., the GNN backbone $\theta_{\text{uni}}$) remain frozen throughout the entire training procedure. Since the adapter has significantly lower capacity than the backbone, it cannot approximating the complex mapping required to cancel the backbone's output into a constant vector. Therefore, the architecture inherently prevents such degenerate solutions.
>
> **Q2**
>
> When facing large graphs, similar to many GNN works, we can perform batched node inference, thus our TWD is computed only between two batched subgraphs. By adjusting the batchsize, the computational cost on large graphs can be effectively constrained. In addition, we have further conducted merging experiments on nine new downstream graph datasets, whose graph sizes exceed 1k nodes and are much larger than molecular graphs. The results also demonstrate the effectiveness of our method. Please refer to the extended Appendix H in the revised paper (marked in blue).
>
> **Q3**
>
> Without the transport constraints, TWD reduces to the original Wasserstein Distance. Here, we conducted an experiment testing the performance variations between TWD and WD in process of node embedding alignment as follow:
>
> | method | tox21   | toxcast | sider   | clintox | bbbp    | bace    | hiv     | muv     |
> | --------------- | ------- | ------- | ------- | ------- | ------- | ------- | ------- | ------- |
> | **TWD**               | 77.3826 | 65.7579 | 64.3668 | 74.1025 | 67.0718 | 86.5241 | 74.0936 | 81.3726 |
> | **WD**              | 76.6997 | 65.1437 | 62.6778 | 74.4142 | 65.6539 | 81.6032 | 74.1059 | 79.5371 |
>
> From the results, we observe that TWD loss consistently outperforms WD loss on the majority of tasks.
> We believe this is because TWD explicitly encodes graph topology, especially local neighborhood structure, providing an inductive bias that guides the alignment of node embeddings in a structurally meaningful manner.
>
> Regarding the suggestion that giving more freedom to the transport plan may be beneficial: while standard WD allows unrestricted transport between any pair of nodes, this flexibility can be harmful in the GNN setting. Without topology constraints, the optimal transport plan may match nodes that are structurally unrelated, potentially introducing noise into the embedding alignment. In contrast, TWD's topology-aware constraint effectively regularizes the transport plan, focusing alignment on meaningful, structurally relevant node pairs. Therefore, the apparent freedom in WD does not translate into improved downstream performance and may even degrade it, as confirmed by the experimental results.

---

> ### Author Response · Authors · 2025-11-20
> **Response to Question 4 & 5**
>
> **Q4**
>
> We appreciate the reviewer's question regarding the practicality of assuming multiple fine-tuned GNNs. In our view, this scenario is also common in the graph learning community. For example, public resources such as GROVER (https://github.com/tencent-ailab/grover) provide GNN checkpoints fine-tuned on BBBP, BACE, SIDER, ClinTox, Tox21, and other MoleculeNet datasets. Many GNN works in the pretrain-finetune paradigm [1][2][3] also release well-trained checkpoints publicly.
>
> In real drug-discovery workflows, different teams often fine-tune the same pretrained GNN on distinct assays, resulting in multiple task-specific models that must be maintained and deployed. Our method provides a practical post-hoc consolidation step: it merges these existing models into a single multi-task GNN with lightweight adapter training, thereby reducing deployment and maintenance costs. We believe similar scenarios are also common in other graph domains.
>
> Thus, the graph model merging scenario is realistic, and our framework directly addresses this practical need.
>
>
> [1] Hu, Weihua, et al. Strategies for pre-training graph neural networks. arXiv preprint arXiv:1905.12265 (2019).
> [2] Jiezhong Qiu, Qibin Chen, Yuxiao Dong, Jing Zhang, Hongxia Yang, Ming Ding, Kuansan Wang, and Jie Tang. Gcc: Graph contrastive coding for graph neural network pre-training. In Proceedings of the 26th ACM SIGKDD international conference on knowledge discovery & data mining, pages 1150–1160, 2020.
> [3] Hu, Z., Dong, Y., Wang, K., Chang, K. W., & Sun, Y. (2020, August). Gpt-gnn: Generative pre-training of graph neural networks. In Proceedings of the 26th ACM SIGKDD international conference on knowledge discovery & data mining (pp. 1857-1867).
>
> **Q5**
>
> Thanks for the suggession. We have added the standard deviation to the main experimental table, following the original results (marked in blue in the revised paper).

---

> > ### Comment · Reviewer_tomN · 2025-11-21
> > **Answer to rebuttal**
> >
> > I thank the authors for addressing all my questions.
> >
> > - Q1: I'm not sure I agree with this argument. For (1) wouldn't smoothing neighborhoods eventually lead to global smoothing (this is exactly the oversmoothing scenario. For (2) I think this would need to be addressed experimentally as it is not possible to know exactly whether a given adapter network has a "low enough" capacity. In any case the results for the method are positive so it seems that in practice this is not an issue.
> >
> > - Q2, Q3 I thank the authors for the additional results.
> >
> > - Q5 I thank the authors for adding standard deviation
> >
> > I think all my concerns have been addressed and I am happy to raise my score

---

> > > ### Author Response · Authors · 2025-11-21
> > >
> > > We sincerely thank the reviewer for the thoughtful follow-up and for the willingness to raise the score. We truly appreciate the constructive feedback and the time spent evaluating our work.

---

### Official Review · Reviewer_YDHz · 2025-10-31

**Soundness:** 3
**Presentation:** 3
**Contribution:** 2
**Rating:** 4
**Confidence:** 4

**Summary:**

This work primarily proposes a graph model fusion framework that first performs coarse fusion of graph models using task-specific operations, and then trains task adapters via TWD loss, leveraging an MoE-based routing mechanism to reduce conflicts after merging and enhance knowledge sharing.

**Strengths:**

1. Originality.The authors' work is original.

2. Quality. The quality of the authors' work is excellent; the assumptions and experiments mutually support each other.

3. Clarity. The authors' work is clear. In particular, the figures fully explain what they are doing.

4. Significance. Given the well-known characteristics of graph data, obtaining a unified cross-modal model in the graph domain is an important problem.

**Weaknesses:**

1. Data Selection: The data selection in this work is confusing to me—why do the experiments only use graph-level datasets and not discuss more widely adopted node-level benchmarks (e.g., Cora)?

2. Lack of Discussion on Related Works: The paper lacks sufficient discussion of related work, which makes it hard to follow. To my understanding, the authors’ method can be summarized as parameter merging combined with knowledge distillation (please correct me if I’m wrong). Even within the graph learning community, this type of approach has been extensively explored—for instance, ParetoGNN [1] (during training) and WAS [2] (during fine-tuning). Could you please clarify how your design differs from or builds upon these prior works?

---

[1] ICLR '23, https://arxiv.org/abs/2210.02016
[2] ICLR '24, https://arxiv.org/abs/2403.01400

**Questions:**

please refer to weaknesses.

Overall, I think this is a good paper, but there are some unclear aspects that prevent me from fully evaluating its contribution. If the authors could clarify my concerns, I would be happy to raise my score.

---

> ### Author Response · Authors · 2025-11-20
> **Response to Weaknesses**
>
> **W1**
>
> Thanks for your significant suggestion!
>
> As in previous work on computer vision and natural language processing [1,2], model merging is typically performed under the pretrain-finetune paradigm. Based on this, we aim to apply graph model merging in a similar setting.
> Molecular property prediction is the usually widely adopted downstream task in the pretrain-finetune setting for GNNs and graph data [3,4]. Notably, many existing studies on graph finetuning methods also focus on these molecular datasets, thus we choose the dataset same as [3,4] in this paper.
>
> We have now further conducted experiments on nine new downstream tasks including both node-level and graph-level, which previously used in GCC [5]. These nine task datasets span domains such as transportation networks, academic collaboration graphs, movie and entertainment networks, and social network graphs. We reused the pretrained GNN provided by GCC [5], and fine-tuned GNN for each task. Then we merge the nine fine-tuned models using G-Merging and baseline methods. We report the F1-score on each task as shown below.
>
> | Method              | usa_airport | europe_airport | brazil_airport | h-index   | imdb-binary | imdb-multi | collab    | rdt-b     | rdt-5k    |
> | ------------------- | ------------ | ----- | ------------ | --------- | ----------- | ---------- | ------ | --------- | --------- |
> | **Pre-trained**     | 41.68        | 37.85           | 56.54           | 56.84     | 59.60       | 39.27      | 61.44     | 59.70     | 32.07     |
> | **Weight Average**  | 56.49        | 56.32           | 45.09           | 65.29     | 60.40       | 39.88      | 62.12     | 61.32     | 33.09     |
> | **Task Arithmetic**  | 58.20        | **58.31**       | 42.90           | 69.10     | 59.80       | 42.60      | 60.80     | 67.80     | 27.80     |
> | **G-Merging**       | **59.01**    | 57.24           | **60.52**       | **72.87** | **61.29**   | **47.30**  | **62.22** | **70.57** | **39.11** |
>
> Among the tasks in the table above, the first four are node classification tasks, and the last five are graph classification tasks.
> The results demonstrate that G-Merging is robust and can generalize effectively to node-level tasks and other graph domains.
>
> We have added the details of this experiment (setup, datasets description, and analysis) in Appendix H of the revised paper (highlighted in blue).
>
>
> [1] Prateek Yadav, Derek Tam, Leshem Choshen, Colin A Raffel, and Mohit Bansal. Ties-merging: Resolving interference when merging models. Advances in Neural Information Processing Systems, 36:7093–7115, 2023.
> [2] Enneng Yang, Zhenyi Wang, Li Shen, Shiwei Liu, Guibing Guo, Xingwei Wang, and Dacheng Tao. Adamerging: Adaptive model merging for multi-task learning. arXiv preprint arXiv:2310.02575, 2023.
> [3] WANG Zhili, DI Shimin, CHEN Lei, and ZHOU Xiaofang. Search to fine-tune pre-trained graph neural networks for graph-level tasks. In 2024 IEEE 40th International Conference on Data Engineering (ICDE), pages 2805–2819. IEEE, 2024.
> [4] Hu, Weihua, et al. Strategies for pre-training graph neural networks. arXiv preprint arXiv:1905.12265 (2019).
> [5] Jiezhong Qiu, Qibin Chen, Yuxiao Dong, Jing Zhang, Hongxia Yang, Ming Ding, Kuansan Wang, and Jie Tang. Gcc: Graph contrastive coding for graph neural network pre-training. In Proceedings of the 26th ACM SIGKDD international conference on knowledge discovery & data mining, pages 1150–1160, 2020.
>
>
>
>
> **W2**
>
> Thanks for your feedback. We would like to clarify that our method is based on the setting where a set of finetuned GNNs for downstream tasks is available, and it differs substantially from the two papers you cited. Our method is a post-training method, whereas ParetoGNN [1] is a pretraining method and WAS [2] is a finetuning method. Here we summarize the core differences in the settings of these methods:
>
> **G-Merging(ours)**: Given $K$ downstream tasks, $K$ fine-tuned GNNs obtained from a shared pretrained checkpoint, and an unlabeled graph, the method aims to merging them into a unified multi-task GNN that can handle all downstream tasks.
>
> **ParetoGNN**[1]: Given a set of pretext graph pretraining tasks and graph data for pretraining, the method aims to learn a well-pretrained GNN with strong generalization capability via dynamically reconciling the task-loss weights.
>
> **WAS**[2]: Given a set of pretrained GNNs trained on different graph pretraining tasks and labeled graph data for a specific downstream task, the method aims to obtain a fine-tuned GNN model for that downstream task.
>
> Overall, these above indicate that G-Merging addresses a fundamentally different problem setting from ParetoGNN [1] and WAS [2], and to the best of our knowledge, this is the first work on graph model merging.
>
> [1] ICLR '23, https://arxiv.org/abs/2210.02016
> [2] ICLR '24, https://arxiv.org/abs/2403.01400

---

> > ### Comment · Reviewer_YDHz · 2025-11-24
> >
> > Thank you for your response. My main concerns have been addressed, so I will adjust my score to 6. And I still recommend that the authors include these clarifications (particularly the additional experiments) in the main exp section to help readers better understand the meaning of ``merging`` throughout the work.

---

> > > ### Author Response · Authors · 2025-11-24
> > >
> > > Thank you for your thoughtful feedback and for reconsidering your score. We appreciate your suggestion to include the clarifications, particularly the additional experiments, in the main experimental section. We will revise the manuscript accordingly to highlight these aspects more prominently, which will provide readers with a clearer understanding of the merging process throughout the work.
> > >
> > > Thank you again for your valuable input.

---

### Official Review · Reviewer_J2df · 2025-10-31

**Soundness:** 2
**Presentation:** 3
**Contribution:** 2
**Rating:** 6
**Confidence:** 4

**Summary:**

The paper introduces G-Merging, a three-phase framework to merge multiple task-specific GNNs into a single multi-task model. Phase I performs task arithmetic to coarsely combine fine-tuned models into a unified encoder. Phase II trains lightweight adapters (node- and graph-level) on each task to align the unified model’s representations to those of the original fine-tuned models via a Topology-aware Wasserstein Distance (TWD) plus an L1 graph-level loss. Phase III composes all task adapters into a training-free MoE with a topology-aware router that computes per-input soft weights from representation similarities, enabling test-time expert selection. Across eight molecular property prediction datasets with GIN/GCN backbones and two pretraining strategies, G-Merging matches or exceeds many weight-merging baselines and approaches per-task fine-tuned upper bounds while using far fewer trainable parameters during adapter training.

**Strengths:**

1. The TWD loss constrains optimal transport by the input graph’s adjacency, encouraging local smoothness in node embeddings and addressing structural heterogeneity that undermines naïve merging. This is a principled graph-specific twist on WD and is well-motivated by prior observations of task/domain clustering.

2. The three phases, task arithmetic (shared knowledge), adapter training (exclusive knowledge), and train-free MoE routing, are clearly specified with equations and an algorithmic sketch, making the method easy to reimplement or ablate.

3. While being, to my knowledge, the first study of model merging in the graph domain is noteworthy, the paper should better justify why such a method is necessary. Given how long model merging has been around, its absence in the graph domain likely reflects factors beyond the task difficulty, for example a lack of clear necessity.

**Weaknesses:**

1. This method still appears computationally intensive, because a task-specific adapter must be trained for each model involved in the merging. If the computational cost of this training is comparable to, or not much lower than, training a GNN from scratch, it seems to run counter to the original purpose of merging. This concern rests on two points: first, GNN training is itself quite lightweight; second, merging is intended to reduce the training burden when facing new tasks. In addition, how transferable are the trained adapters? Do we need to retrain a new set of adapters for every configuration (number/combination of models)?

2. Compared with the baselines, this method appears to require more GPU memory at inference. The baselines’ memory footprint is essentially that of a single model, whereas this approach involves an MoE component, which requires a router to decide whether and how each adapter is applied during inference. Moreover, the results in Tables 1 and 2 suggest that introducing the MoE architecture has little to no positive effect on overall performance and seems largely dispensable.

3. Figure 3(b) is only shown in the main text but is barely discussed in the experimental description. The paper mentions on lines 473–473 that “as the set of selected experts becomes larger, indicating access to more diverse knowledge, the model performance improves accordingly.” However, this statement does not match the results in Figure 3(b): as the set of selected experts grows, performance first drops and then rises, rather than monotonically increasing. In the appendix, Figure 7 presents more fine-grained results but still lacks detailed analysis. I recommend that the authors provide a thorough analysis of this experiment. Based on Figure 7, the Top-K strategy appears to have markedly different effects across datasets. How do the authors balance these differences in practice, and what explains these divergent impacts?

4. It seems that all tasks are molecular property prediction. It is unclear whether TWD and routing generalize to non-molecule domains (social, knowledge, large sparse graphs). Please add some non-molecular datasets or discuss constraints.

**Questions:**

See weakness. I'll adjust my rating according to the authors' rebuttal.

---

> ### Author Response · Authors · 2025-11-20
> **Response to Weakness 1 & 2**
>
> Thank you for the valuable and constructive feedback. Below we provide our responses.
>
> **W1**
>
> We would like to clarify that the training phase of our method is lightweight. In the paper, we have included experiments and analysis regarding the efficiency of our approach. Specifically, we recorded the runtime of the entire G-Merging process and compared it with a multi-task learning baseline, where a multi-task GNN is trained from scratch on all tasks jointly. The results are presented in Table 7 (Appendix E), and we also summarize the main results below:
>
> | Method                  | Average score | Time cost       |
> |-------------------------|------------------------------|------------------|
> | **Multi-Task Learning** | 71.2                         | 144 min 45 s     |
> | **G-Merging**           | 73.9                         | 57 min 56 s      |
>
> Compared to training a multi-task GNN from scratch, our G-Merging method significantly reduces the training time. This is expected, as the adapter module contains only a very small number of parameters relative to the full GNN (see Table 4 in Section 5) and converges more easily during training. Therefore, G-Merging serves as an efficient way to construct a multi-task GNN, which not counters the purpose of model merging.
>
> Moreover, standard multi-task training may not balance the tasks well, potentially leading to suboptimal performance compared to G-Merging. This is also a reason for using G-Merging.
>
> In addition, since the adapters are task-specific, we consider they can only generalize to similar tasks. This is also the underlying motivation for our MoE-based router module, as discussed in Section 5.2 (Router Analysis). When facing new tasks, it is not necessary to retrain all adapters. Instead, we only need to train new adapters for each new task independently. The training of task-specific adapters does not interfere with existing ones, maintaining efficiency and flexibility.
>
> **W2**
>
> Our adapter modules are lightweight (with the number of parameters shown in Table 4, Section 5), so the additional inference cost is still acceptable. Furthermore, advanced multi-task approaches that aim for better performance often employ more sophisticated architectures beyond a single unified model (e.g., the Twin-Merging baseline in our paper). To quantify this, we conduct experiments to record the peak GPU memory usage for each task during inference using both the G-Merging model and the baseline models. The results are shown below (in MB, measured on a single RTX 4090 GPU).
>
> | method | tox21   | toxcast | sider   | clintox | bbbp    | bace    | hiv     | muv     |
> | --------------- | ------- | ------- | ------- | ------- | ------- | ------- | ------- | ------- |
> | **A single model**       | 547.96  | 698.93  | 86.46  | 168.25 | 242.74 | 231.79 | 116.93 | 289.66 |
> | **G-Merging**             | 940.96  | 1049.77 | 121.16  | 306.93  | 405.84  | 258.14  | 127.07  | 357.50  |
> | **Twin-Merging** | 1226.57 | 1248.24 | 224.2207  | 353.934  | 440.59  | 368.25  | 235.05  | 378.41  |
>
> We observe that the GPU memory usage of G-Merging isn't significantly more than that of a single model and is lower than that of the baseline method, Twin-Merging. Considering the superior performance of G-Merging, we believe that this level of GPU memory requirement is acceptable.
>
> Regarding the reviewer's concern about the contribution of the MoE component, although its effect on the overall performance is not dominant, we believe it remains meaningful for several reasons:
> (1) Our MoE construction is entirely train-free, which enhances the robustness of the merged model without introducing additional training cost.
> (2) The MoE layer enables complementary knowledge sharing across tasks. For complex datasets such as MUV, where a single task-specific adapter may be insufficient, the MoE mechanism yields markedly improved performance.
> (3) Together with our top-k selection experiments, when the number of tasks is large, we observe that the benefits of MoE become more pronounced as the number of experts increases. This indicates that its advantages may manifest more strongly in heterogeneous or large task collections, which we leave as a promising direction for future work.

---

> ### Author Response · Authors · 2025-11-20
> **Response to Weakness 3**
>
> **W3**
>
> Thank you for your suggestion. We provide a detailed explanation of this experiment below and have added it to both the main text and the Appendix (marked in blue in the revised paper).
>
> Top-k selection means that the MoE router sorts the assigned expert weights in descending order, keeps the top k weights, and sets all other weights to zero. Formally, let $\mathbf{w} = \{w_ 1,w_ 2,...,w_ K\}$ denotes the original MoE weights for K experts, the top-k selected weights are defined as :
> $$
> \mathbf{w}^{\text{top-}k}_ i =
> \begin{cases}
> w_ i, & \text{if } w_i \text{ is among the top-$k$ largest weights in } \mathbf{w},  \\\\0, & \text{otherwise},
> \end{cases}
> \quad i = 1, \dots, K
> $$
> Then we compute the final MoE weights as $\mathbf{w}^{\text{final}} = \text{softmax}\{\mathbf{w}^{\text{top-}k}_ i\}$ .
>
> For this experiment, we use the GIN (contextpred) pretrained model, and keep all other setting same as in the main experiments.
>
> The overall results show that as the number k of selected experts increases from 1 to 8, the model performance first decreases and then increases. When relatively few task experts are selected, the source task cannot receive sufficient support from other tasks, and due to the heterogeneity of adapters, the effectiveness of the original adapter may also be affected, leading to performance degradation. In contrast, when more task experts are selected, the router can better fulfill its role by assigning appropriate balancing weights, allowing a large pool of similar task experts to collaboratively support the source task, thereby improving the performance.
> For the fine-grained results in Appendix Figure 7, we attribute the varying effects of the top-k strategy across tasks to their intrinsic characteristics. Overall, most datasets (such as ToxCast, SIDER, BACE, and ClinTox) exhibit the trend described above, where performance first decreases and then increases as k grows. For some datasets (such as BBBP and HIV), this pattern is less pronounced, likely because these tasks share fewer similar counterparts within the task set. In practice, we do not adopt the top-k mechanism in our MoE module, i.e., all experts remain active during inference.

---

> ### Author Response · Authors · 2025-11-20
> **Response to Weakness 4**
>
> **W4**
>
> Thanks for your significant suggestion!
>
> As in previous work on computer vision and natural language processing [1,2], model merging is typically performed under the pretrain-finetune paradigm.
> Molecular property prediction is the usually widely adopted downstream task in the pretrain-finetune setting for GNNs and graph data [3,4]. Notably, many existing studies on graph finetuning methods also focus on molecular datasets, thus we choose the dataset same as [3,4] in this paper.
>
> We have now further conducted experiments on nine new downstream tasks beyond the molecule graph domain, which previously used in GCC [5]. These nine task datasets span domains such as transportation networks, academic collaboration graphs, movie and entertainment networks, and social network graphs. We reused the pretrained GNN provided by GCC [5], and fine-tuned GNN for each task. Then we merge the nine fine-tuned models using G-Merging and baseline methods. We report the F1-score on each task as shown below.
>
> | Method              | usa_airport | europe_airport | brazil_airport | h-index   | imdb-binary | imdb-multi | collab    | rdt-b     | rdt-5k    |
> | ------------------- | ------------ | ----- | ------------ | --------- | ----------- | ---------- | ------ | --------- | --------- |
> | **Pre-trained**     | 41.68        | 37.85           | 56.54           | 56.84     | 59.60       | 39.27      | 61.44     | 59.70     | 32.07     |
> | **Weight Average**  | 56.49        | 56.32           | 45.09           | 65.29     | 60.40       | 39.88      | 62.12     | 61.32     | 33.09     |
> | **Task Arithmetic**  | 58.20        | **58.31**       | 42.90           | 69.10     | 59.80       | 42.60      | 60.80     | 67.80     | 27.80     |
> | **G-Merging**       | **59.01**    | 57.24           | **60.52**       | **72.87** | **61.29**   | **47.30**  | **62.22** | **70.57** | **39.11** |
>
> The results demonstrate that G-Merging is not restricted to molecular property prediction and can generalize effectively to non-molecular domains such as social networks and larger graph structures. We have added the details of this experiment (setup, datasets description, and analysis) in Appendix H of the revised paper (highlighted in blue).
>
>
> [1] Prateek Yadav, Derek Tam, Leshem Choshen, Colin A Raffel, and Mohit Bansal. Ties-merging: Resolving interference when merging models. Advances in Neural Information Processing Systems, 36:7093–7115, 2023.
> [2] Enneng Yang, Zhenyi Wang, Li Shen, Shiwei Liu, Guibing Guo, Xingwei Wang, and Dacheng Tao. Adamerging: Adaptive model merging for multi-task learning. arXiv preprint arXiv:2310.02575, 2023.
> [3] WANG Zhili, DI Shimin, CHEN Lei, and ZHOU Xiaofang. Search to fine-tune pre-trained graph neural networks for graph-level tasks. In 2024 IEEE 40th International Conference on Data Engineering (ICDE), pages 2805–2819. IEEE, 2024.
> [4] Hu, Weihua, et al. Strategies for pre-training graph neural networks. arXiv preprint arXiv:1905.12265 (2019).
> [5] Jiezhong Qiu, Qibin Chen, Yuxiao Dong, Jing Zhang, Hongxia Yang, Ming Ding, Kuansan Wang, and Jie Tang. Gcc: Graph contrastive coding for graph neural network pre-training. In Proceedings of the 26th ACM SIGKDD international conference on knowledge discovery & data mining, pages 1150–1160, 2020.

---

> > ### Comment · Reviewer_J2df · 2025-11-25
> >
> > Thanks for the authors' detailed response. Most of my concerns are addressed. I will keep my positive score unchanged for the acceptance of the paper.

---

> > > ### Author Response · Authors · 2025-11-25
> > >
> > > We sincerely thank you for acknowledging our work and for the positive score. We truly appreciate the constructive feedback and the time spent reviewing our work.

---

### Author Response · Authors · 2025-12-03
**Rebuttal Summary**

Dear AC, SAC, PCs, and Reviewers,

We are aware that the recent incident has impacted the ICLR community, and we sincerely regret the situation. We thank the PCs for the timely actions and the new protocol, and we appreciate the newly assigned AC for taking on the additional workload. We also thank all reviewers for their valuable and constructive feedback, as well as their positive engagement during the rebuttal, including the score increase.

To reduce the AC's heavy workload under the new assignment, we provide a concise summary of the main reviewer concerns and our corresponding responses as follows, including additional experiments and newly added analyses in the revised manuscript.

### **Rebuttal Summary:**

First, all reviewers acknowledge the novelty of our work, the clarity of our method, and the meaningful contribution of introducing the first systematic model merging framework for GNNs. We have provided detailed responses to all the weaknesses and questions raised, and we believe that most of them have been addressed. Below, we summarize the major concerns from the initial reviews along with our responses.

**Reviewer J2df  Score: 6 (responsed and maintained the score)**

- Main concerns:
(1) Training cost and inference memory; (2) limited analysis of Top-k expert selection; (3) potential lack of generalization beyond molecular graphs.

- Response:
(1) We added experiments to record the training time and GPU memory usage. The results show that G-Merging trains 2.5× faster than multi-task training from scratch, because adapters are lightweight. Moreover, GPU memory usage is only moderately higher than a single model and lower than Twin-Merging, while providing stronger performance.
(2) We added detailed analysis for the Top-k behavior (now included in both main text and Appendix).
(3) We added experminets on nine new downstream tasks across heterogeneous graph domains (transportation, collaboration, social, movie networks), demonstrating clear generalization beyond molecules.

**Reviewer YDHz  Score: 4 (responsed and raised score to 6)**

- Main concerns:
(1) Dataset selection (lack of node-level benchmarks);
(2) Relation to ParetoGNN[1] and WAS[2].

- Response:
(1) We added nine new tasks from GCC, including four node-level and five graph-level datasets, showing robust performance across settings.
(2) We clarified that G-Merging is fundamentally a post-training model merging method, while ParetoGNN[1] is a pretraining method and WAS[2] a finetuning method; the problem settings are completely different.

**Reviewer tomN  Score: 6 (responsed and raised score to 8)**

- Main concerns:
(1) Whether TWD encourages oversmoothing;
(2) Effect of removing topology constraints;
(3) Practicality of assuming multiple fine-tuned GNNs.

- Response:
(1) we respond that TWD does not encourage oversmoothing because constraints limit transport to local neighborhoods, and adapters are low-capacity and cannot collapse embeddings.
(2) We added expriments showing that Removing topology constraints (i.e., using standard WD) consistently worsens performance across all tasks.
(3) We cite several realistic scenarios (e.g., GROVER[3] checkpoints and drug-discovery workflows) where merging multiple fine-tuned GNNs can reduce deployment costs.

**Reviewer VpK5  Score: 6 (responsed and maintained the score)**

- Main concerns:
(1) Support for tasks with different output dimensions;
(2) Hyperparameter sensitivity;
(3) Limited task diversity;


- Response:
(1) Tasks already have different output dimensions; only graph encoders are merged and classification heads are kept per-task.
(2) Sensitivity analysis shows stable performance; key hyperparameters in our main merging experiments were not tuned.
(3) We added experiments on nine diverse graph tasks (node & graph level), showing strong generalization.


In the newly submitted manuscript, we have added three parts:
- Experiments on nine additional types of graph and domain tasks
- Discription and analysis of top-k experiments
- Standard deviations for all main results

and the revised text or newly added sections are marked in blue.

We hope this rebuttal summary helps to reduce the AC's workload and facilitates a more thorough evaluation of our paper. We also sincerely thank the ACs, SAC, PCs, and all reviewers for their efforts and dedication in handling the recent incident.

Best regards, The Authors

[1] ICLR '23, https://arxiv.org/abs/2210.02016
[2] ICLR '24, https://arxiv.org/abs/2403.01400
[3] https://github.com/tencent-ailab/grover

---

### Meta-Review · Area_Chair_AHs4 · 2025-12-28

**Summary:**

In this paper, the authors propose a graph model merging framework for multi-task fine-tuned GNNs, incorporating task arithmetic, lightweight task adapters, MoE architecture, etc. Experimental results show the effectiveness of the proposed method.

In general, the reviewers acknowledge that this is a novel problem and that the proposed method is technically sound. Some concerns were raised regarding insufficient experimental comparisons as well as details regarding method designs. The authors have provided extensive discussions, including additional experiments, in the rebuttal. The reviewers generally acknowledge that their concerns have been (at least partially) addressed. Overall, the paper presents a solid method for an under-explored problem, and clearly passes the bar for acceptance.

**Reviewer Concerns:**

Most main concerns regarding experiments and detailed explanations regarding the proposed method seem to have been addressed.

**Reviewer Scores:**

For Reviewer J2df, the initial rating is 6, and it is likely to stay at 6.

For Reviewer YDHz, the initial rating is 4, and it is likely to increase to 6.

For Reviewer tomN, the initial rating is 6, and it is likely to increase to 8.

For Reviewer VpK5, the initial rating is 6, and it is likely to stay at 6

---

### Decision · Program_Chairs · 2026-01-26

Accept (Poster)